# Simultaneously Learning Stochastic and Adversarial Episodic MDPs with Known Transition

**Tiancheng Jin**
University of Southern California
tiancheng.jin@usc.edu

**Haipeng Luo**
University of Southern California
haipengl@usc.edu

## Abstract

This work studies the problem of learning episodic Markov Decision Processes with known transition and bandit feedback. We develop the first algorithm with a "best-of-both-worlds" guarantee: it achieves $\mathcal{O}(\log T)$ regret when the losses are stochastic, and simultaneously enjoys worst-case robustness with $\widetilde{\mathcal{O}}(\sqrt{T})$ regret even when the losses are adversarial, where $T$ is the number of episodes. More generally, it achieves $\widetilde{\mathcal{O}}(\sqrt{C})$ regret in an intermediate setting where the losses are corrupted by a total amount of $C$. Our algorithm is based on the Follow-the-Regularized-Leader method from Zimin and Neu [26], with a novel hybrid regularizer inspired by recent works of Zimmert et al. [27, 29] for the special case of multi-armed bandits. Crucially, our regularizer admits a non-diagonal Hessian with a highly complicated inverse. Analyzing such a regularizer and deriving a particular self-bounding regret guarantee is our key technical contribution and might be of independent interest.

## 1 Introduction

We study the problem of learning episodic Markov Decision Processes (MDPs). In this problem, a learner interacts with the environment through $T$ episodes. In each episode, the learner starts from a fixed state, then sequentially selects one of the available actions and transits to the next state according to a fixed transition function for a fixed number of steps. The learner observes only the visited states and the loss for each visited state-action pair, and her goal is to minimize her regret, the difference between her total loss over $T$ episodes and that of the optimal fixed policy in hindsight.

When the losses are adversarial and can change arbitrarily between episodes, the state-of-the-art is achieved by the UOB-REPS algorithm of [12] with near-optimal regret $\widetilde{\mathcal{O}}(\sqrt{T})$ (ignoring dependence on other parameters). On the other hand, the majority of the literature focuses on the stochastic/i.i.d. loss setting where the loss for each state-action pair follows a fixed distribution. For example, Azar et al. [5] achieve the minimax regret $\widetilde{\mathcal{O}}(\sqrt{T})$ in this case. Moreover, the recent work of Simchowitz and Jamieson [23] shows the first non-asymptotic gap-dependent regret bound of order $\mathcal{O}(\log T)$ for this problem, which is considerably more favorable than the worst-case $\widetilde{\mathcal{O}}(\sqrt{T})$ regret.

A natural question then arises: *is it possible to achieve the best of both worlds with one single algorithm?* In other words, can we achieve $\mathcal{O}(\log T)$ regret when the losses are stochastic, and *simultaneously* enjoy worst-case robustness with $\widetilde{\mathcal{O}}(\sqrt{T})$ regret when the losses are adversarial? Considering that the existing algorithms from [12] and [23] for these two settings are drastically different, it is highly unclear whether this is possible.

In this work, we answer the question affirmatively and develop the first algorithm with such a best-of-both-worlds guarantee, under the condition that the transition function is known. We emphasize that even in the case with known transition, the problem is still highly challenging. For example,

the adversarial case was studied in [26] and still requires using the Follow-the-Regularized-Leader (FTRL) or Online Mirror Descent framework from the online learning literature (see e.g., [10]) over the *occupancy measure* space, which the UOB-REPS algorithm [12] adopts as well. This is still significantly different from the algorithms designed for the stochastic setting.

Moreover, our algorithm achieves the logarithmic regret $\mathcal{O}(\log T)$ for a much broader range of situations besides the usual stochastic setting. In fact, neither independence nor identical distributions are required, as long as a certain gap condition similar to that of [27] (for multi-armed bandits) holds (see Eq. (2)). Even more generally, our algorithm achieves $\widetilde{\mathcal{O}}(\log T + \sqrt{C})$ regret in an intermediate setting where the losses are corrupted by a total amount of $C$. This bound smoothly interpolates between the logarithmic regret for the stochastic setting and the worst-case $\widetilde{\mathcal{O}}(\sqrt{T})$ regret for the adversarial setting as $C$ increases from 0 to $T$.

**Techniques.**    Our algorithm is mainly inspired by recent advances in achieving best-of-both-worlds guarantees for the special case of multi-armed bandits or semi-bandits [24, 27, 29]. These works show that, perhaps surprisingly, such guarantees can be obtained with the standard FTRL framework originally designed only for the adversarial case. All we need is a carefully designed regularizer and a particular analysis that relies on proving a certain kind of *self-bounding regret bounds*, which then automatically implies $\mathcal{O}(\log T)$ regret for the stochastic setting, and more generally $\widetilde{\mathcal{O}}(\sqrt{C})$ regret for the case with $C$ corruption.

We greatly extend this idea to the case of learning episodic MDPs. As mentioned, Zimin and Neu [26] already solved the adversarial case using FTRL over the occupancy measure space, in particular with Shannon entropy as the regularizer in the form $\sum_{s,a} q(s,a) \ln q(s,a)$, where $q(s,a)$ is the occupancy for state $s$ and action $a$. Our key algorithmic contribution is to design a new regularizer based on the 1/2-Tsallis-entropy used in [27]. However, we argue that using only the Tsallis entropy, in the form of $-\sum_{s,a} \sqrt{q(s,a)}$, is not enough. Instead, inspired by the work of [29] for semi-bandits, we propose to use a *hybrid* regularizer in the form $-\sum_{s,a} (\sqrt{q(s,a)} + \sqrt{q(s) - q(s,a)})$ where $q(s) = \sum_a q(s,a)$. In fact, to stabilize the algorithm, we also need to add yet another regularizer in the form $-\sum_{s,a} \log q(s,a)$ (known as log-barrier), borrowing the idea from [7, 8, 16]. See Section 2.2 and Section 3 for more detailed discussions on the design of our regularizer.

More importantly, we emphasize that analyzing our new regularizer requires significantly new ideas, mainly because it admits a *non-diagonal Hessian* with a highly complicated inverse. Indeed, the key of the FTRL analysis lies in analyzing the quadratic norm of the loss estimator with respect to the inverse Hessian of the regularizer. As far as we know, almost all regularizers used in existing FTRL methods are decomposable over coordinates and thus admit a diagonal Hessian, making the analysis relatively straightforward (with some exceptions mentioned in related work below). Our approach is the first to apply and analyze an explicit non-decomposable regularizer with non-diagonal Hessian. Our analysis heavily relies on rewriting $q(s)$ in a different way and constructing the Hessian inverse recursively (see Section 4). The way we analyze our algorithm and derive a self-bounding regret bound for MDPs is the key technical contribution of this work and might be of independent interest.

While we only resolve the problem with known transition, we believe that our approach, providing the first best-of-both-worlds result for MDPs, sheds light on how to solve the general case with unknown transition.

**Related work.**    We refer the reader to [23] for earlier works on gap-dependent logarithmic regret bounds for learning MDPs with stochastic losses, and to [12] for earlier works on learning MDPs with adversarial losses. Using very different techniques, the recent work of Lykouris et al. [18] also develops an algorithm for the stochastic setting that is robust to a certain amount of adversarial corruption to the environment (including both the transition and the losses). Their algorithm does not ensure a worst-case bound of $\mathcal{O}(\sqrt{T})$ and can only tolerate $o(\sqrt{T})$ amount of corruption, while our algorithm ensures $\mathcal{O}(\sqrt{T})$ regret always. On the other hand, their algorithm works even under unknown transition while ours cannot.

For the special case of multi-armed bandits (essentially our setting with one single state), the question of achieving best-of-both-worlds was first considered by Bubeck and Slivkins [6]. Since then, different improvements have been proposed over the years [22, 4, 21, 24, 17, 9, 27, 29]. As mentioned, our regularizer is largely based on the most recent advances from [27, 29], briefly reviewed in Section 2.2.

As far as we know, all existing works on using FTRL with a regularizer that admits a non-diagonal Hessian do not require calculating the Hessian inverse explicitly. For example, the SCRiBLe algorithm of [1, 2] for efficient bandit linear optimization and its variants (e.g., [20, 11]) use any self-concordant barrier [19] of the decision set as the regularizer, and the entire analysis only relies on certain properties of self-concordant barriers and does not even require knowing the explicit form of the regularizer. As another example, when applying FTRL over a space of matrices, the regularizer usually also does not decompose over the entries (see e.g., [14]). However, the Hessian inverse is often well-known from matrix calculus already. These previous attempts are all very different from our analysis where we need to explicitly work out the inverse of the non-diagonal Hessian.

## 2 Preliminaries

The problem of learning an episodic MDP through $T$ episodes is defined by a tuple $(S, A, P, L, \{\ell_t\}_{t=1}^T)$, where $S$ and $A$ are the finite state and action space respectively, $P :$ $S \times A \times S \to [0, 1]$ is the transition function such that $P(s'|s, a)$ is the probability of moving to state $s'$ after executing action $a$ at state $s$, $L$ is the number of interactions within each episode, and $\ell_t : S \times A \to [0, 1]$ is the loss function for episode $t$, specifying the loss of each state-action pair.

Without loss of generality (see detailed discussions in [12]), the MDP is assumed to have the following layered structure. First, the state space $S$ consists of $L + 1$ layers $S_0, \ldots, S_L$ such that $S = \bigcup_{k=0}^{L} S_k$ and $S_i \cap S_j = \emptyset$ for $i \neq j$. Second, $S_0$ and $S_L$ are singletons, containing only the start state $s_0$ and the terminal state $s_L$ respectively. Third, transitions are only possible between consecutive layers. In other words, if $P(s'|s, a) > 0$, then $s' \in S_{k+1}$ and $x \in S_k$ for some $k$.

Ahead of time, the environment decides $(S, A, P, L, \{\ell_t\}_{t=1}^T)$, with $S, A, L$, and $P$ revealed to the learner. The interaction between the learner and the environment then proceeds in $T$ episodes. For each episode $t$, the learner decides a stochastic policy $\pi_t : S \times A \to [0, 1]$, where $\pi_t(a|s)$ is the probability of selecting action $a$ at state $s$, then executes the policy starting from the initial state $s_0$ until reaching the final state $s_L$, yielding and observing a sequence of $L$ state-action-loss tuples $(s_0, a_0, \ell_t(s_0, a_0)), \ldots, (s_{L-1}, a_{L-1}, \ell_t(s_{L-1}, a_{L-1}))$, where action $a_k$ is drawn from $\pi_t(\cdot|s_k)$ and state $s_{k+1}$ is drawn from $P(\cdot|s_k, a_k)$, for each $k = 0, \ldots, L - 1$. Importantly, the learner only observes the loss of the visited action-state pairs and nothing else about the loss function $\ell_t$, known as the *bandit feedback* setting.

For any policy $\pi$, with slight abuse of notation we denote its expected loss in episode $t$ as $\ell_t(\pi) = \mathbb{E}\big[\sum_{k=0}^{L-1} \ell_t(s_k, a_k)\big]$, where the state-action pairs $(s_0, a_0), \ldots, (s_{L-1}, a_{L-1})$ are generated according to the transition function $P$ and the policy $\pi$. The expected regret of the learner with respect to a policy $\pi$ is defined as $\text{Reg}_T(\pi) = \mathbb{E}\big[\sum_{t=1}^{T} \ell_t(\pi_t) - \sum_{t=1}^{T} \ell_t(\pi)\big]$, which is the difference between the total loss of the learner and that of policy $\pi$. The goal of the leaner is to minimize her regret with respect to an optimal policy, denoted as $\text{Reg}_T = \max_\pi \text{Reg}_T(\pi)$. Throughout the paper, we use $\mathring{\pi} : S \to A$ to denote a *deterministic* optimal policy, which is known to always exist.

**Occupancy measures.** To solve the problem using techniques from online learning, we need the concept of "occupancy measures" proposed in [26]. Specifically, fixing the MDP of interest, every stochastic policy $\pi$ induces an occupancy measure $q^\pi : S \times A \to [0, 1]$ with $q^\pi(s, a)$ being the probability of visiting state-action pair $(s, a)$ by following policy $\pi$. With this concept, the expected loss of a policy $\pi$ in episode $t$ can then be written as a simple linear function of $q^\pi$: $\ell_t(\pi) = \sum_{s \neq s_L, a \in A} q^\pi(s, a)\ell_t(s, a)$, which we denote as $\langle q^\pi, \ell_t \rangle$, and the regret can be written as $\text{Reg}_T = \mathbb{E}\big[\sum_{t=1}^{T} \langle q_t - \mathring{q}, \ell_t \rangle\big]$ where $q_t = q^{\pi_t}$ and $\mathring{q} = q^{\mathring{\pi}}$.

In other words, the problem essentially becomes an instance of online linear optimization with bandit feedback, where in each episode the learner proposes an occupancy measure $q_t \in \Omega$. Here, $\Omega$ is the set of all possible occupancy measures and is known to be a polytope satisfying two constraints [26]: first, for every $k = 0, \ldots, L - 1$, $\sum_{s \in S_k} \sum_{a \in A} q(s, a) = 1$; second, for every $k = 1, \ldots, L - 1$ and every state $s \in S_k$,

$$\sum_{s' \in S_{k-1}} \sum_{a' \in A} q(s', a')P(s|s', a') = \sum_a q(s, a). \tag{1}$$

Having an occupancy measure $q_t \in \Omega$, one can directly find its induced policy via $\pi_t(a|s) = q_t(s, a)/\sum_{a' \in A} q_t(s, a')$.

**More notation.** We use $k(s)$ to denote the index of the layer to which state $s$ belongs, and $\mathbb{I}\{\cdot\}$ to denote the indicator function whose value is $1$ if the input holds true and $0$ otherwise. For a positive definite matrix $M \in \mathbb{R}^{K \times K}$, $\|x\|_M \triangleq \sqrt{x^\top M x}$ is the quadratic norm of $x \in \mathbb{R}^K$ with respect to $M$. Throughout the paper, we use $\widetilde{\mathcal{O}}(\cdot)$ to hide terms of order $\log T$.

## 2.1 Conditions on loss functions

So far we have not discussed how the environment decides the loss functions $\ell_1, \ldots, \ell_T$. We consider two general settings. The first one is the *adversarial setting*, where there is no assumption at all on how the losses are generated — they can even be generated in a malicious way after seeing the learner's algorithm (but not her random seeds).[1] The O-REPS algorithm of [26] achieves $\text{Reg}_T = \widetilde{\mathcal{O}}(\sqrt{L|S||A|T})$ in this case, which was shown to be optimal.

The second general setting we consider subsumes many cases such as the stochastic case. Generalizing [28] (the full version of [27]) for bandits, we propose to summarize this setting by the following condition on the losses: there exist a gap function $\Delta : S \times A \to \mathbb{R}_+$, a mapping $\pi^\star : S \to A$, and a constant $C \geq 0$ such that

$$\text{Reg}_T = \mathbb{E}\left[\sum_{t=1}^{T} \langle q_t - \mathring{q}, \ell_t \rangle\right] \geq \mathbb{E}\left[\sum_{t=1}^{T} \sum_{s \neq s_L} \sum_{a \neq \pi^\star(s)} q_t(s, a)\Delta(s, a)\right] - C \tag{2}$$

holds for all sequences of $q_1, \ldots, q_T$. While seemingly strong and hard to interpret, this condition in fact subsumes many existing settings as explained below.

**Stochastic losses.** In the standard stochastic setting studied by most works in the literature, $\ell_1, \ldots, \ell_T$ are i.i.d. samples of a fixed and unknown distribution. In this case, by the performance difference lemma (see e.g., [13, Lemma 5.2.1]), Eq. (2) is satisfied with $\pi^\star = \mathring{\pi}$, $C = 0$, and $\Delta(s, a) = Q(s, a) - \min_{a' \neq a} Q(s, a')$ where $Q$ is the Q function of the optimal policy $\mathring{\pi}$.[2] In fact, Eq. (2) holds with equality in this case. Here, $\Delta(s, a)$ is the sub-optimality gap of action $a$ at state $s$, and plays a key role in the optimal logarithmic regret for learning MDPs as shown in [23].

**Stochastic losses with corruption.** More generally, consider a setting where the environment first generates $\ell'_1, \ldots, \ell'_T$ as i.i.d. samples of an unknown distribution (call the corresponding MDP $\mathcal{M}$), and then corrupt them in an arbitrary way to arrive at the final loss functions $\ell_1, \ldots, \ell_T$. Then Eq. (2) is satisfied with $\pi^\star$ being the optimal policy of $\mathcal{M}$, $C = 2\sum_{t=1}^{T} \sum_{k<L} \max_{s \in S_k, a} |\ell_t(s, a) - \ell'_t(s, a)|$ being the total amount of corruption, and $\Delta(s, a) = Q(s, a) - \min_{a' \neq a} Q(s, a')$ where $Q$ is the Q function of policy $\pi^\star$ with respect to $\mathcal{M}$. This is because: $\text{Reg}_T \geq \sum_{t=1}^{T} \langle q_t - q^{\pi^\star}, \ell_t \rangle = \langle q_t - q^{\pi^\star}, \ell'_t \rangle + \langle q_t, \ell_t - \ell'_t \rangle - \langle q^{\pi^\star}, \ell_t - \ell'_t \rangle \geq \sum_{t=1}^{T} \sum_{s \neq s_L} \sum_{a \neq \pi^\star(s)} q_t(s, a)\Delta(s, a) - C$, where in the last step we use the performance difference lemma again and the definition of $C$.

Compared to the corruption model studied by [18], the setting considered here is more general in the sense that $C$ measures the total amount of corruption in the loss functions, instead of the number of corrupted episodes as in [18]. On the other hand, they allow corrupted transition as well when an episode is corrupted, while our transition is always fixed and known.

## 2.2 Follow-the-Regularized-Leader and self-bounding regret

FTRL is one of the standard frameworks to derive online learning algorithms for adversarial environments. In our context, FTRL computes $q_t = \arg\min_{q \in \Omega} \sum_{\tau < t} \langle q, \widehat{\ell_\tau} \rangle + \psi_t(q)$ where $\widehat{\ell_\tau}$ is some estimator for the loss function $\ell_\tau$ and $\psi_t$ is a regularizer usually of the form $\psi_t = \frac{1}{\eta_t}\psi$ for some learning rate $\eta_t > 0$ and a fixed convex function $\psi$. The O-REPS algorithm exactly falls into this

framework with $\widehat{\ell}_t$ being the importance-weighted estimator (more details in Section 3) and $\psi$ being the entropy function.

While traditionally designed for adversarial environments, somewhat surprisingly FTRL was recently shown to be able to adapt to stochastic environments with $\mathcal{O}(\log T)$ regret as well, in the context of multi-armed bandits starting from the work by Wei and Luo [24], which was later greatly improved by Zimmert and Seldin [27, 28]. This approach is conceptually extremely clean and only relies on designing a regularizer that enables a certain kind of self-bounding regret bound. We briefly review the key ideas below since our approach is largely based on extending the same idea to MDPs.

Multi-armed bandit is a special case of our setting with $L = 1$. Thus, the concept of states does not play a role, and below we write $q(s_0, a)$ as $q(a)$ for conciseness. The regularizer used in [27] is the $1/2$-Tsallis-entropy, originally proposed in [3], and is defined as $\psi(q) = -\sum_a \sqrt{q(a)}$. With a simple learning rate schedule $\eta_t = 1/\sqrt{t}$, it was shown that FTRL ensures the following adaptive regret bound for some constant $B > 0$,

$$\text{Reg}_T \leq \mathbb{E}\left[ B \sum_{t=1}^{T} \sum_{a \neq a^\star} \sqrt{\frac{q_t(a)}{t}} \right] \tag{3}$$

where $a^\star$ can be *any* action. The claim is now that, solely based on Eq. (3), one can derive the best-of-both-worlds guarantee already, without even further considering the details of the algorithm. To see this, first note that applying Cauchy-Schwarz inequality ($\sum_a \sqrt{q_t(s)} \leq \sqrt{|A|}$) immediately leads to a worst-case robustness guarantee of $\text{Reg}_T = \mathcal{O}(\sqrt{|A|T})$ (optimal for multi-armed bandits).

More importantly, suppose now Condition (2) holds (note again that there is only one state $s = s_0$ and we write $\Delta(s_0, a)$ as $\Delta(a)$). Then picking $a^\star = \pi^\star(s_0)$, one can arrive at the following self-bounding regret using Eq. (3):

$$\text{Reg}_T \leq \mathbb{E}\left[ \sum_{t=1}^{T} \sum_{a \neq a^\star} \frac{q_t(a)\Delta(a)}{2z} + \frac{zB^2}{2t\Delta(a)} \right] \leq \frac{\text{Reg}_T + C}{2z} + zB^2 \sum_{a \neq a^\star} \frac{\log T}{\Delta(a)},$$

where the first step uses the AM-GM inequality and holds for any $z > 1/2$, and the second step uses Eq. (2) and the fact $\sum_{t=1}^{T} 1/t \leq 2 \log T$. Note that we have bounded the regret in terms of itself (hence the name self-bounding). Rearranging then gives $\text{Reg}_T \leq \frac{2z^2 B^2}{2z-1} \sum_{a \neq a^\star} \frac{\log T}{\Delta(a)} + \frac{C}{2z-1}$. It just remains to pick the optimal $z$ to minimize the bound. Specifically, using the shorthand $U = \frac{1}{2}B^2 \sum_{a \neq a^\star} \frac{\log T}{\Delta(a)}$ and $x = 2z - 1 > 0$, the bound can be simplified to $\text{Reg}_T \leq 2U + Ux + (C + U)/x$, and finally picking the best $x$ to balance the last two terms gives $\text{Reg}_T \leq 2U + 2\sqrt{U(C + U)} \leq 4U + 2\sqrt{UC} = 2B^2 \sum_{a \neq a^\star} \frac{\log T}{\Delta(a)} + \sqrt{2CB^2 \sum_{a \neq a^\star} \frac{\log T}{\Delta(a)}}$.

In the case with stochastic losses ($C = 0$), the final bound is $\mathcal{O}\left( \sum_{a \neq a^\star} \frac{\log T}{\Delta(a)} \right)$, exactly matching the lower bound for stochastic multi-armed bandits [15]. More generally, in the corruption model, the regret is order $\mathcal{O}(\log T + \sqrt{C \log T})$, smoothly interpolating between the bounds for the stochastic setting and the adversarial setting as $C$ increases from 0 to $T$.

Finally, we remark that although not mentioned explicitly, the follow-up work [29] reveals that using a *hybrid* regularizer in the form $\psi(q) = -\sum_a (\sqrt{q(a)} + \sqrt{1 - q(a)})$, with Tsallis entropy applied to both $q$ and its complement, also leads to the same bound Eq. (3), via an even simpler analysis. This is crucial for our algorithm design and analysis as explained in the next section.

## 3 Algorithm and Main Results

We are now ready to present our algorithm. Based on the discussions from Section 2.2, our goal is to design an algorithm with a regret bound akin to Eq. (3):

$$\text{Reg}_T \leq \mathbb{E}\left[ B \sum_{t=1}^{T} \sum_{s \neq s_L} \sum_{a \neq \pi(s)} \sqrt{\frac{q_t(s, a)}{t}} \right] \tag{4}$$

---

**Algorithm 1** FTRL with hybrid Tsallis entropy for learning stochastic and adversarial MDPs

---

**Parameters:** $\alpha, \beta, \gamma$
**Define:** hybrid regularizer $\phi_H$ and log-barrier regularizer $\phi_L$ as in Eq. (5) and Eq. (6)
**Define:** valid occupancy measure space $\Omega$ (see Section 2), learning rate $\eta_t = \gamma/\sqrt{t}$
**Initialize:** $\widehat{L}_0(s, a) = 0$ for all $(s, a)$
**for** $t = 1$ **to** $T$ **do**
    compute $q_t = \operatorname{argmin}_{q \in \Omega} \left\langle q, \widehat{L}_{t-1} \right\rangle + \psi_t(q)$ where $\psi_t(q) = \frac{1}{\eta_t}\phi_H(q) + \phi_L(q)$
    execute policy $\pi_t$ where $\pi_t(a|s) = q_t(s, a)/q_t(s)$
    observe $(s_0, a_0, \ell_t(s_0, a_0)), \ldots, (s_{L-1}, a_{L-1}, \ell_t(s_{L-1}, a_{L-1}))$
    construct estimator $\widehat{\ell}_t$ such that: $\forall (s, a), \widehat{\ell}_t(s, a) = \frac{\ell_t(s,a)}{q_t(s,a)}\mathbb{I}\{s_{k(s)} = s, a_{k(s)} = a\}$
    update $\widehat{L}_t = \widehat{L}_{t-1} + \widehat{\ell}_t$

---

for *any* mapping $\pi : S \rightarrow A$. This immediately implies a worst-case regret bound $\text{Reg}_T = \mathcal{O}(\sqrt{L|S||A|T})$ (by applying Cauchy-Schwarz inequality), matching the optimal bound of [26]. Moreover, repeating the same calculation and picking $\pi = \pi^\star$, one can verify that under Condition (2) this leads to a similar bound of order $\mathcal{O}(\log T + \sqrt{C \log T})$ as in the case for multi-armed bandits.

To achieve so, a natural idea is to directly extend the $1/2$-Tsallis-entropy regularizer to MDPs and take $\psi(q) = -\sum_{s,a} \sqrt{q(s,a)}$. However, generalizing the proofs of [27] one can only prove a weaker bound: $\text{Reg}_T \leq B \sum_{t=1}^T \sum_{(s,a) \neq \xi(k(s))} \sqrt{\frac{q_t(s,a)}{t}}$ for any $\xi$ mapping from a layer index to a state-action pair. Compared to the desired Eq. (4), one can see that instead of excluding one arbitrary action $\pi(s)$ for *each* state $s$ in the bound, now we only exclude one arbitrary action for *one single* state specified by $\xi$ in each layer. This is not enough to derive the same results as one can verify.

The hybrid regularizer $-\sum_{s,a}(\sqrt{q(s,a)} + \sqrt{1 - q(s,a)})$ [29] suffers the same issue. However, we propose a natural fix to this hybrid version by replacing 1 with $q(s) \triangleq \sum_a q(s,a)$, that is, the marginal probability of visiting state $s$ under occupancy measure $q$.[3] More concretely, we define (with "$H$" standing for "hybrid")

$$\phi_H(q) = -\sum_{s \neq s_L, a \in A} \left(\sqrt{q(s,a)} + \alpha\sqrt{q(s) - q(s,a)}\right), \quad \text{where} \quad q(s) = \sum_{a \in A} q(s,a) \quad (5)$$

for some parameter $\alpha > 0$, as the key component of our regularizer. Note that in the case of multi-armed bandits, this exactly recovers the original hybrid regularizer since $q(s_0) = 1$ and there is only one state. For MDPs, intuitively each state is dealing with a multi-armed bandit instance, but with total available probability $q(s)$ instead of 1, making $\phi_H$ a natural choice. However, also note another important distinction between $\phi_H$ and the ones discussed earlier: $\phi_H$ does not decompose over the action-state pairs, thus admitting a *non-diagonal Hessian*. This makes the analysis highly challenging since standard FTRL analysis requires analyzing the Hessian inverse of the regularizer. We will come back to this challenge in Section 4.

To further stabilize the algorithm and make sure that $q_t$ and $q_{t+1}$ are not too different (another important requirement of typical FTRL analysis), we apply another regularizer in addition to $\phi_H$, defined as (with "$L$" standing for "log-barrier"):

$$\phi_L(q) = \beta \sum_{s \neq s_L, a \in A} \log \frac{1}{q(s,a)}, \quad (6)$$

for some parameter $\beta > 0$. Our final regularizer for time $t$ is then $\psi_t(q) = \frac{1}{\eta_t}\phi_H(q) + \phi_L(q)$ where $\eta_t = \gamma/\sqrt{t}$ is a decreasing learning rate with parameter $\gamma > 0$. In all our results we pick $\beta = \mathcal{O}(L)$. Thus, the weight for $\phi_L$ is much smaller than that for $\phi_H$. This idea of adding a small amount of log-barrier to stabilize the algorithm was first used in [7] and recently applied in several other works [8, 25, 16]. See more discussions in Section 4.

Our final algorithm is shown in Algorithm 1. In each episode $t$, the algorithm follows standard FTRL and computes $q_t = \operatorname{argmin}_{q \in \Omega} \langle q, \widehat{L}_{t-1} \rangle + \psi_t(q)$ with $\widehat{L}_{t-1} = \sum_{s<t} \widehat{\ell}_s$ being the accumulated estimated loss. Then the policy $\pi_t$ induced from $q_t$ is executed, generating a sequence of state-action-loss tuples. A standard importance-weighted unbiased estimator $\widehat{\ell}_t$ is then constructed with $\widehat{\ell}_t(s,a)$ being the actual loss $\ell_t(s,a)$ divided by $q_t(s,a)$ if the state-action pair $(s,a)$ was visited in this episode, and zero otherwise. Also note that Algorithm 1 can be efficiently implemented since the key FTRL step is a convex optimization problem with $\mathcal{O}(L + |S||A|)$ linear constraints (solving it to an inaccuracy of $\mathcal{O}(1/T)$ is enough clearly).

### 3.1 Main Results

We move on to present the regret guarantees of our algorithm. As mentioned, the goal is to show Eq. (4), and the theorem below shows that our algorithm essentially achieves this (see Appendix A for the proof).

**Theorem 1.** *With $\alpha = 1/\sqrt{|A|}$, $\beta = 64L$, and $\gamma = 1$, Algorithm 1 ensures that $\operatorname{Reg}_T$ is bounded by*

$$\sum_{t=1}^{T} \widetilde{\mathcal{O}} \left( \min \left\{ \mathbb{E} \left[ B \sum_{s \neq s_L} \sum_{a \neq \pi(s)} \sqrt{\frac{q_t(s,a)}{t}} + D \sqrt{\sum_{s \neq s_L} \sum_{a \neq \pi(s)} \frac{q_t(s,a) + \mathring{q}(s,a)}{t}} \right], \frac{D}{\sqrt{t}} \right\} \right) \quad (7)$$

*for any mapping $\pi : S \to A$, where $B = L^{\frac{5}{2}} + L\sqrt{|A|}$ and $D = \sqrt{L|S||A|}$.*

Looking at the first term of the min operator, one sees that we have an extra term compared to the ideal bound Eq. (4). However, this only contributes to small terms in the final bounds as we explain below.[4] Based on this theorem, we next present more concrete regret bounds for our algorithm. First, consider the adversarial setting where there is no further structure in the losses. Simply taking the second term of the min operator in Eq. (7) we obtain the following corollary.

**Corollary 2.** *With $\alpha = 1/\sqrt{|A|}$, $\beta = 64L$, and $\gamma = 1$, the regret of Algorithm 1 is always bounded as*

$$\operatorname{Reg}_T \leq \widetilde{\mathcal{O}} \left( \sqrt{L|S||A|T} \right).$$

Again, this bound matches that of the O-REPS algorithm [26] and is known to be optimal. Note that using the first term of the min operator we can also derive an $\widetilde{\mathcal{O}}(\sqrt{T})$ bound, but the dependence on other parameters would be larger.

On the other hand, under Condition Eq. (2), we obtain the following bound.

**Corollary 3.** *Suppose Condition (2) holds. Then with $\alpha = 1/\sqrt{|A|}$, $\beta = 64L$, and $\gamma = 1$, the regret of Algorithm 1 is bounded as $\operatorname{Reg}_T \leq \mathcal{O}(U + \sqrt{UC})$ where*

$$U = \frac{L|S||A| \log T}{\Delta_{\mathrm{MIN}}} + L^2 \left( L^3 + |A| \right) \sum_{s \neq s_L} \sum_{a \neq \pi^\star(s)} \frac{\log T}{\Delta(s,a)} = \mathcal{O}(\log T)$$

*and $\Delta_{\mathrm{MIN}} = \min_{s \neq s_L, a \neq \pi^\star(s)} \Delta(s,a)$ is the minimal gap. Consequently, in the stochastic setting, we have $\operatorname{Reg}_T = \mathcal{O}(U)$.*

See Appendix A.1 for the proof, whose idea is similar to the discussions in Section 2.2. Note that we are getting an extra term in $U$ involving $1/\Delta_{\mathrm{MIN}}$, which in turn comes from the extra term in Eq. (7) mentioned earlier. For the stochastic setting, the best existing logarithmic bound is $\mathcal{O}\left(L^3 \sum_{s \neq s_L} \sum_{a \neq \pi^\star(s)} \frac{\log T}{\Delta(s,a)}\right)$ from [23] (which also has a dependence on $\frac{1}{\Delta_{\mathrm{MIN}}}$ but is ignored here for simplicity). Our bound has a worse factor $L^2(L^3 + |A|)$. We note that by tuning the parameters $\alpha$ and $\gamma$ differently, one can obtain a better bound in this case. This set of parameters, however, leads to a sub-optimal adversarial bound. See Appendix A.2 for details. Since our goal is to develop one single algorithm that adapts to different environments automatically, we stick to the same set of parameters in the theorem and corollaries above.

As mentioned, in the corruption setting, our bound $\widetilde{\mathcal{O}}(\sqrt{C})$ smoothly interpolates between the logarithmic regret in the no corruption case and the $\widetilde{\mathcal{O}}(\sqrt{T})$ worst-case regret in the adversarial case. On the other hand, the bound from [18] is of order $\widetilde{\mathcal{O}}(C^2)$, only meaningful for $C = o(\sqrt{T})$.

## 4  Analysis Sketch

Analyzing our algorithm requires several new ideas, which we briefly mention in this section through a few steps, with all details deferred to the appendix.

**First step.** While we define $q(s)$ as $\sum_a q(s,a)$, the entire analysis relies on using an equivalent definition of $q(s)$ based on Eq. (1). The benefit of this alternative definition is that it contains the information of the transition function $P$ and implicitly introduces a layered structure to the regularizer, which is important for constructing the Hessian and its inverse recursively. We emphasize that, however, this does *not* change the algorithm at all, because all occupancy measures in $\Omega$ ensure Eq. (1) by definition and the FTRL optimization over $\Omega$ is thus not affected.

**Second step.** Following the standard FTRL analysis one can obtain a regret bound in terms of $\|\widehat{\ell}_t\|_{\nabla^{-2}\phi_H(q_t')}$ for some $q_t'$ between $q_t$ and $q_{t+1}$. Moving from $q_t'$ to $q_t$ is exactly the part where the log-barrier $\phi_L$ is important. Specifically, following the idea of [16, Lemma 9], we prove that $q_t$ and $q_{t+1}$ are sufficiently close, and consequently $\text{Reg}_T$ is mainly bounded by two terms: the penalty term $\sum_{t=1}^T (1/\eta_t - 1/\eta_{t-1})(\phi_H(\mathring{q}) - \phi_H(q_t))$ and the stability term $\sum_{t=1}^T \eta_t\|\widehat{\ell}_t\|^2_{\nabla^{-2}\phi_H(q_t)}$ (see Lemma 5).

**Third step.** Bounding the penalty term already requires a significant departure from the analysis for multi-armed bandits. Specifically, $\phi_H(\mathring{q}) - \phi_H(q_t)$ can be written as

$$\sum_{s \neq s_L} \sqrt{q_t(s)}\,(h_s(\pi_t) - h_s(\mathring{\pi})) + \sum_{s \neq s_L} \left(\sqrt{q_t(s)} - \sqrt{\mathring{q}(s)}\right) h_s(\mathring{\pi}) \tag{8}$$

where $h_s(\pi) = \sum_{a \in A} \sqrt{\pi(a|s)} + \alpha\sqrt{1 - \pi(a|s)}$ is basically the hybrid regularizer for multi-armed bandits (at state $s$) mentioned in Section 2.2. The first term in Eq. (8) can then be bounded as $(1+\alpha)\sum_s \sum_{a \neq \pi(s)} \sqrt{q_t(s,a)}$ for *any* mapping $\pi : S \to A$, in a similar way as in the multi-armed bandit analysis. However, the key difficulty is the second term, which does not appear for multi-armed bandits where there is only one state $s_0$ with $q(s_0) = 1$ for all $q$. This is highly challenging especially because we would like to arrive at a term with a summation over $a \neq \pi(s)$ as in the first term. Our main idea is to separately bound $\sqrt{q_t(s)} - \sqrt{q^\pi(s)}$ and $\sqrt{q^\pi(s)} - \sqrt{\mathring{q}(s)}$ via a key induction lemma (Lemma 18) that connects them to similar terms in previous layers. We remark that this term is the source of the extra term $\sqrt{\sum_{s \neq s_L} \sum_{a \neq \pi(s)}(q_t(s,a) + \mathring{q}(s,a))/t}$ in our main regret bound Eq. (7). The complete proof for bounding the penalty term is in Appendix C.

**Fourth step.** Analyzing the stability term is yet another highly challenging part, since it requires working out the Hessian inverse $\nabla^{-2}\phi_H(q_t)$. The high-level idea of our proof is to first write the Hessian in a recursive form based on the layered structure introduced by writing $q(s)$ differently as mentioned in the first step. Then we apply Woodbury matrix identity to obtain a recursive form of the Hessian inverse. Finally, we argue that only certain parts of the Hessian inverse matter, and these parts enjoy nice properties allowing us to eventually bound $\mathbb{E}\left[\|\widehat{\ell}_t\|^2_{\nabla^{-2}\phi_H(q_t)}\right]$ by $8eL^2\left(\sqrt{L} + 1/\alpha L\right)\sum_{s \neq s_L}\sum_{a \neq \pi(s)}\sqrt{q_t(s,a)}$, again, for any mapping $\pi$. See Appendix D for the complete proof.

## 5  Conclusion

In this work, we provide the first algorithm for learning episodic MDPs that automatically adapts to different environments with favorable guarantees. Our algorithm applies a natural regularizer with a complicated non-diagonal Hessian to the FTRL framework, and our analysis for obtaining

a self-bounding regret bound requires several novel ideas. Apart from improving our bound in Corollary 3, one key future direction is to remove the known transition assumption, which appears to demand new techniques since it is hard to also control the bias introduced by estimating the transition in terms of the adaptive bound in Eq. (7).

## Broader Impact

This work is mostly theoretical, with no negative outcomes. Researchers working on theoretical aspects of online learning, bandit problems, and reinforcement learning (RL) may benefit from our results. Although our algorithm deals with the tabular setting and is not directly applicable to common RL applications with a large state and action space, it sheds light on how to increase robustness of a learning algorithm while adapting to specific instances, and serves as an important step towards developing more practical, adaptive, and robust RL algorithms, which in the long run might find its applications in the real world.

## Acknowledgments and Disclosure of Funding

We thank Max Simchowitz for many helpful discussions. HL is supported by NSF Awards IIS-1755781 and IIS-1943607, and a Google Faculty Research Award.

## Footnotes

[1]Technically, this is a setting with an *oblivious* adversary, where $\ell_t$ does not depend on the learner's previous actions. However, this is only for simplicity and our results generalize to adaptive adversaries directly.

[2]One caveat here is that since we require $\Delta(s, a)$ to be non-zero, the optimal action for each state needs to be unique for Eq. (2) to hold. The work of [28] requires this uniqueness condition for multi-armed bandits as well, which is conjectured to be only an artifact of the analysis.

[3]We find it intriguing that while the Tsallis entropy regularizer and its hybrid version work equally well for multi-armed bandits, only the latter admits a natural way to be generalized to learning MDPs.

[4]The fact that this form of weaker bounds is also sufficient for the self-bounding property might be of interest for other bandit problems as well.

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
