[Supplementary Material]

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

# A  Proof of Theorem 1 and Corollary 3

In this section, we provide the proof (outline) of Theorem 1, the proof of Corollary 3 (Appendix A.1), discussions on parameter tuning (Appendix A.2), and also some preliminaries on the Hessian of our regularizer which are useful for the rest of the appendix (Appendix A.3).

We prove the following version of Theorem 1 with general value of the parameters $\alpha$ and $\gamma$, which facilitates further discussions on parameter tuning. Taking $\alpha = 1/\sqrt{|A|}$ and $\gamma = 1$ clearly recovers Theorem 1.

**Theorem 4.** *With $\beta = 64L$, Algorithm 1 guarantees:*

$$\text{Reg}_T = \mathcal{O}\left(\sum_{t=1}^{T}\min\left\{\mathbb{E}\left[X\sum_{s\neq s_L}\sum_{a\neq\pi(s)}\sqrt{\frac{q_t(s,a)}{t}} + Y\sqrt{\sum_{s\neq s_L}\sum_{a\neq\pi(s)}\frac{q_t(s,a)+\mathring{q}(s,a)}{t}}\right], \frac{Z}{\sqrt{t}}\right\}\right)$$
$$+ \mathcal{O}\left(L|S||A|\log T\right)$$

*for any mapping $\pi : S \to A$, where coefficients $X$, $Y$ and $Z$ are defined as:*

$$X = \frac{1+\alpha}{\gamma} + \gamma L^2\left(\sqrt{L} + \frac{1}{\alpha L}\right), Y = \frac{\sqrt{L|S|}\,(1+\alpha|A|)}{\gamma}, Z = \sqrt{L|S||A|}\left(\frac{1+\alpha\sqrt{|A|}}{\gamma} + \gamma\right).$$

The proof of this theorem relies on the following three important lemmas, which respectively correspond to the second, third, and fourth step of the analysis sketch in Section 4 and are proven in Appendix B, Appendix C, and Appendix D. The first one decomposes the regret into two terms (penalty and stability), with an additional small term of order $\mathcal{O}(L|S||A|\log T)$. The second one bounds the penalty term, while the third one bounds the stability term.

**Lemma 5** (Regret decomposition). *With $\beta = 64L$, Algorithm 1 ensures:*

$$\text{Reg}_T \leq \underbrace{\sum_{t=1}^{T}\left(\frac{1}{\eta_t} - \frac{1}{\eta_{t-1}}\right)\mathbb{E}\left[\phi_H(\mathring{q}) - \phi_H(q_t)\right]}_{penalty} + \underbrace{8\sum_{t=1}^{T}\eta_t\mathbb{E}\left[\left\|\widehat{\ell}_t\right\|_{\nabla^{-2}\phi_H(q_t)}^2\right]}_{stability}$$
$$+ \mathcal{O}\left(L|S||A|\log T\right).$$

**Lemma 6** (Penalty). *The hybrid regularizer $\phi_H$ defined in Eq. (5) ensures that $\phi_H(\mathring{q}) - \phi_H(q_t)$ is bounded by*

$$(1+\alpha)\sum_{s\neq s_L}\sum_{a\neq\pi(a)}\sqrt{q_t(s,a)} + (1+\alpha|A|)\sqrt{|S|L}\min\left\{1, 2\sqrt{\sum_{s\neq s_L}\sum_{a\neq\pi(s)}q_t(s,a)+\mathring{q}(s,a)}\right\},$$

*for all $t = 1, \ldots, T$, where $\pi$ can be any mapping from $S$ to $A$.*

**Lemma 7** (Stability). *Algorithm 1 guarantees that $\mathbb{E}\left[\|\widehat{\ell}_t\|_{\nabla^{-2}\phi_H(q_t)}^2\right]$ is bounded by*

$$\min\left\{4\sqrt{L|S||A|}, \mathbb{E}\left[8eL^2\left(\sqrt{L} + \frac{1}{\alpha\cdot L}\right)\sum_{s\neq s_L}\sum_{a\neq\pi(a)}\sqrt{q_t(s,a)}\right]\right\},$$

*for all $t = 1, \ldots, T$, where $\pi$ can be any mapping from $S$ to $A$.*

*Proof of Theorem 4.* For notational convenience, we denote

$$J_1(t) = \sqrt{\frac{1}{t}}, J_2(t) = \mathbb{E}\left[\sum_{s\neq s_L}\sum_{a\neq\pi(a)}\sqrt{\frac{q_t(s,a)}{t}}\right], J_3(t) = \mathbb{E}\left[\sqrt{\sum_{s\neq s_L}\sum_{a\neq\pi(s)}\frac{q_t(s,a)+\mathring{q}(s,a)}{t}}\right].$$

By Lemma 6 and the fact $1/\eta_t - 1/\eta_{t-1} = (\sqrt{t} - \sqrt{t-1})/\gamma = \frac{1}{\gamma(\sqrt{t}+\sqrt{t-1})} \le 1/(\gamma\sqrt{t})$, the penalty term can be bounded by

$$\mathcal{O}\left(\sum_{t=1}^{T} \mathbb{E}\left[C_1 \sum_{s \ne s_L} \sum_{a \ne \pi(a)} \sqrt{\frac{q_t(s,a)}{t}}\right] + C_2 \min\left\{\frac{1}{\sqrt{t}}, \mathbb{E}\left[\sqrt{\sum_{s \ne s_L} \sum_{a \ne \pi(s)} \frac{q_t(s,a) + \mathring{q}(s,a)}{t}}\right]\right\}\right),$$

that is, $\mathcal{O}\left(\sum_{t=1}^{T} C_1 J_2(t) + C_2 \min\{J_1(t), J_3(t)\}\right)$, where $C_1 = \frac{1+\alpha}{\gamma}$ and $C_2 = \frac{(1+\alpha|A|)\sqrt{|S|L}}{\gamma}$.

On the other hand, by Lemma 7, the stability term is bounded by

$$\mathcal{O}\left(\sum_{t=1}^{T} \min\left\{\frac{C_3}{\sqrt{t}}, \mathbb{E}\left[C_4 \sum_{s \ne s_L} \sum_{a \ne \pi(a)} \sqrt{\frac{q_t(s,a)}{t}}\right]\right\}\right),$$

that is, $\mathcal{O}\left(\sum_{t=1}^{T} \min\{C_3 J_1(t), C_4 J_2(t)\}\right)$, where $C_3 = \gamma\sqrt{L|S||A|}$ and $C_4 = \gamma L^2\left(\sqrt{L} + 1/\alpha L\right)$.

Finally, we plug these bounds into Lemma 5 and show that

$$\text{Reg}_T = \mathcal{O}\left(\sum_{t=1}^{T} C_1 J_2(t) + C_2 \min\{J_1(t), J_3(t)\} + \min\{C_3 J_1(t), C_4 J_2(t)\}\right)$$
$$+ \mathcal{O}\left(L|S||A|\log T\right).$$

Noticing that $J_2(t) = \sum_{s \ne s_L} \sum_{a \ne \pi(a)} \sqrt{\frac{q_t(s,a)}{t}} \le \sqrt{\frac{L|S||A|}{t}} = \sqrt{L|S||A|} J_1(t)$ by Cauchy-Schwarz inequality, we further have

$$\text{Reg}_T = \mathcal{O}\left(\sum_{t=1}^{T} C_1 J_2(t) + C_2 \min\{J_1(t), J_3(t)\} + \min\{C_3 J_1(t), C_4 J_2(t)\}\right)$$
$$+ \mathcal{O}\left(L|S||A|\log T\right)$$
$$= \mathcal{O}\left(\sum_{t=1}^{T} \min\left\{C_1\sqrt{L|S||A|} J_1(t), C_1 J_2(t)\right\} + \min\{(C_2+C_3) J_1(t), C_2 J_3(t) + C_4 J_2(t)\}\right)$$
$$+ \mathcal{O}\left(L|S||A|\log T\right)$$
$$= \mathcal{O}\left(\sum_{t=1}^{T} \min\left\{\left(C_1\sqrt{L|S||A|} + C_2 + C_3\right) J_1(t), C_2 J_3(t) + (C_1+C_4) J_2(t)\right\}\right)$$
$$+ \mathcal{O}\left(L|S||A|\log T\right).$$

Therefore, we prove the regret bound stated in Theorem 4 with $X = C_1 + C_4$, $Y = C_2$, and $Z = C_1\sqrt{L|S||A|} + C_2 + C_3 = \mathcal{O}\left(\sqrt{L|S||A|}\left(\frac{1+\alpha\sqrt{|A|}}{\gamma} + \gamma\right)\right)$. In particular, setting $\gamma = 1$ and $\alpha = 1/\sqrt{|A|}$ exactly leads to Theorem 1. $\qquad\square$

## A.1    Proof of Corollary 3

The proof mostly follows the discussions in Section 2.2, except that we need to deal with the extra term involving $\mathring{q}$. To do that, we first introduce the following important implication of Condition (2).

**Lemma 8.** *Suppose Condition* (2) *holds. Then the optimal occupancy measure $\mathring{q}$ ensures*

$$T \sum_{s \ne s_L} \sum_{a \ne \pi^\star(s)} \mathring{q}(s,a)\Delta(s,a) \le C.$$

*Proof.* Simply setting $q_1 = \cdots = q_T = \mathring{q}$ in Condition (2) gives

$$0 = \mathbb{E}\left[\sum_{t=1}^{T}\langle\mathring{q} - \mathring{q}, \ell_t\rangle\right] \geq T\sum_{s\neq s_L}\sum_{a\neq\pi^\star(s)}\mathring{q}(s,a)\Delta(s,a) - C,$$

and rearranging finishes the proof. □

*Proof of Corollary 3.* By Theorem 4 (setting $\alpha = 1/\sqrt{|A|}$ and $\gamma = 1$ as stated in Corollary 3 and picking $\pi = \pi^\star$), $\mathrm{Reg}_T$ is bounded by

$$\kappa\left(L|S||A|\log T + \sum_{t=1}^{T}\mathbb{E}\left[B\sum_{s\neq s_L}\sum_{a\neq\pi^\star(s)}\sqrt{\frac{q_t(s,a)}{t}} + D\sqrt{\sum_{s\neq s_L}\sum_{a\neq\pi^\star(s)}\frac{q_t(s,a) + \mathring{q}(s,a)}{t}}\right]\right)$$

where $\kappa$ is a constant, $B = L^{\frac{5}{2}} + L\sqrt{|A|}$ and $D = \sqrt{L|S||A|}$.

For any $z > 0$, we have

$$\kappa\sum_{t=1}^{T}\mathbb{E}\left[B\sum_{s\neq s_L}\sum_{a\neq\pi^\star(s)}\sqrt{\frac{q_t(s,a)}{t}}\right]$$

$$\leq \sum_{t=1}^{T}\mathbb{E}\left[\sum_{s\neq s_L}\sum_{a\neq\pi^\star(s)}\sqrt{\frac{q_t(s,a)\Delta(s,a)}{z}\cdot\frac{z\kappa^2 B^2}{t\Delta(s,a)}}\right]$$

$$\leq \mathbb{E}\left[\sum_{t=1}^{T}\sum_{s\neq s_L}\sum_{a\neq\pi^\star(s)}\left(\frac{q_t(s,a)\Delta(s,a)}{2z} + \frac{z\kappa^2 B^2}{2t\Delta(s,a)}\right)\right]$$

$$\leq \frac{\mathrm{Reg}_T + C}{2z} + z\kappa^2 L^2(L^3 + |A|)\sum_{s\neq s_L}\sum_{a\neq\pi^\star(s)}\frac{\log T}{\Delta(s,a)}$$

where the third line uses the AM-GM inequality, and the last line uses Eq. (2) and the fact $\sum_{t=1}^{T} 1/t \leq 2\log T$.

For the other term, we have for any $z > 0$:

$$\kappa\sum_{t=1}^{T}\mathbb{E}\left[D\sqrt{\sum_{s\neq s_L}\sum_{a\neq\pi^\star(s)}\frac{q_t(s,a) + \mathring{q}(s,a)}{t}}\right]$$

$$= \sum_{t=1}^{T}\mathbb{E}\left[\sqrt{\left(\sum_{s\neq s_L}\sum_{a\neq\pi^\star(s)}\frac{(q_t(s,a) + \mathring{q}(s,a))\Delta_{\mathrm{MIN}}}{z}\right)\cdot\frac{\kappa^2 D^2}{t\Delta_{\mathrm{MIN}}}}\right]$$

$$\leq \mathbb{E}\left[\sum_{t=1}^{T}\left(\sum_{s\neq s_L}\sum_{a\neq\pi^\star(s)}\frac{(q_t(s,a) + \mathring{q}(s,a))\Delta_{\mathrm{MIN}}}{2z}\right) + \frac{z\kappa^2 D^2}{2t\Delta_{\mathrm{MIN}}}\right]$$

$$\leq \mathbb{E}\left[\sum_{t=1}^{T}\left(\sum_{s\neq s_L}\sum_{a\neq\pi^\star(s)}\frac{(q_t(s,a) + \mathring{q}(s,a))\Delta(s,a)}{2z}\right) + \frac{z\kappa^2 D^2}{2t\Delta_{\mathrm{MIN}}}\right]$$

$$\leq \frac{\mathrm{Reg}_T + 2C}{2z} + \frac{z\kappa^2 L|S||A|\log T}{\Delta_{\mathrm{MIN}}}$$

where the third line uses the AM-GM inequality again, the third line uses the definition of $\Delta_{\mathrm{MIN}}$, and the last line uses Eq. (2), Lemma 8, and the fact $\sum_{t=1}^{T} 1/t \leq 2\log T$.

Combining the inequalities, we have

$$\mathrm{Reg}_T \leq \frac{\mathrm{Reg}_T}{z} + \frac{2C}{z} + z\kappa^2 U + \kappa V$$

where we use the shorthand $U$ (already defined in the statement of Corollary 3) and $V$ as

$$U = L^2(L^3 + |A|) \sum_{s \neq s_L} \sum_{a \neq \pi^\star(s)} \frac{\log T}{\Delta(s,a)} + \frac{L|S||A|\log T}{\Delta_{\text{MIN}}}, \quad V = L|S||A|\log T.$$

For any $z > 1$, we can further rearrange and arrive at

$$\begin{aligned}
\text{Reg}_T &\leq \frac{2}{(z-1)}C + \frac{z^2}{z-1}\kappa^2 U + \frac{z}{z-1}\kappa V \\
&= \frac{2}{x}C + \frac{(x+1)^2}{x}\kappa^2 U + \frac{x+1}{x}\kappa V \\
&= \frac{1}{x}\left(2C + \kappa V + \kappa^2 U\right) + x\left(\kappa^2 U\right) + \left(2\kappa^2 U + \kappa V\right)
\end{aligned}$$

where we define $x = z - 1 > 0$ and replace all $z$'s in the second line. Picking the optimal $x$ to balance the first terms gives

$$\begin{aligned}
\text{Reg}_T &\leq 2\sqrt{\left(2C + \kappa V + \kappa^2 U\right)\left(\kappa^2 U\right)} + 2\kappa^2 U + \kappa V \\
&\leq 2\kappa\sqrt{2UC} + 2\sqrt{\kappa^3 UV} + 4\kappa^2 U + \kappa V \\
&\leq 2\kappa\sqrt{2UC} + \sqrt{\kappa^3}\left(U + V\right) + 4\kappa^2 U + \kappa V \\
&\leq \mathcal{O}\left(U + V + \sqrt{UC}\right)
\end{aligned}$$

where the second line follows from the fact $\sqrt{a+b} \leq \sqrt{a} + \sqrt{b}$, and the third line uses AM-GM inequality. Finally, noticing that $V \leq \frac{L|S||A|\log T}{\Delta_{\text{MIN}}} \leq U$ finishes the proof. $\qquad\square$

## A.2 Different tuning for the stochastic case

Here, we consider the case with stochastic losses (or more generally the case where Condition (2) holds with $C = 0$), and point out what the best bound one can get by tuning $\alpha$ and $\gamma$ optimally. For simplicity, we consider the worst case when $\Delta(s,a) = \Delta_{\text{MIN}}$ holds for all state-action pairs $(s,a)$ with $a \neq \pi^\star(s)$. Repeating the same argument in the proof of Corollary 3 with the general bound from Theorem 4, one can verify that the final bound is

$$\text{Reg}_T \leq \mathcal{O}\left(\left[\frac{1+\alpha}{\gamma} + \gamma L^2\left(\sqrt{L} + \frac{1}{\alpha L}\right)\right]^2 \frac{|S||A|\log T}{\Delta_{\text{MIN}}} + \left(\frac{(1+\alpha|A|)\sqrt{|S|L}}{\gamma}\right)^2 \frac{\log T}{\Delta_{\text{MIN}}}\right)$$
$$+ \mathcal{O}\left(L|S||A|\log T\right).$$

Picking the optimal parameters leads to

$$\text{Reg}_T \leq \mathcal{O}\left(|S|\sqrt{|A|}L^{3/2}\left(\sqrt{|A|}L + L^{3/2} + |A| + A^{3/4}\sqrt{L}\right)\frac{\log T}{\Delta_{\text{MIN}}} + L|S||A|\log T\right).$$

This is better than the bound stated in Corollary 3, and could even be better than the bound $\mathcal{O}\left(\frac{L^3|S||A|\log T}{\Delta_{\text{MIN}}}\right)$ achieved by StrongEuler [23] (although they consider a harder setting where the transition function is unknown). However, this set of parameters leads to a sub-optimal bound for the adversarial case unfortunately.

## A.3 The Hessian of $\phi_H$

We calculate the Hessian of our hybrid regularizer $\phi_H$ in this section, which is important for the analysis in Appendix B and Appendix D. As mentioned in Section 4 (first step), one important trick we do is to use a different but equivalent definition of $q(s)$. Specifically, we analyze the following regularizer:

$$\phi_H(q) = -\sum_{s \neq s_L, a \in A}\left(\sqrt{q(s,a)} + \alpha\sqrt{q(s) - q(s,a)}\right),$$

$$\text{where } q(s) = \begin{cases} \sum_{s' \in S_{k(s)-1}} \sum_{a' \in A} q(s',a')P(s|s',a'), & \text{if } k(s) \neq 0, \\ 1, & \text{else.} \end{cases} \tag{9}$$

We emphasize again that, within the feasible set $\Omega$, this definition is exactly equivalent to Eq. (5), and thus we are not changing the algorithm at all.

**Lemma 9.** *The Hessian of the regularizer $\phi_H(q)$ defined in Eq. (9) is specified by the following:*

- *for any $k = 1, \ldots, L-1$, $s' \in S_{k-1}$, $s \in S_k$, and $a, a' \in A$,*

$$\frac{\partial^2 \phi_H}{\partial q(s', a') \partial q(s, a)} = \frac{-\alpha P(s|s', a')}{4 \left(q(s) - q(s, a)\right)^{3/2}}; \tag{10}$$

- *for any $k = 1, \ldots, L-1$, $s, s' \in S_{k-1}$, and $(s, a) \neq (s', a')$,*

$$\frac{\partial^2 \phi_H}{\partial q(s', a') \partial q(s, a)} = \sum_{s'' \in S_k} \sum_{a'' \in A} \frac{\alpha P(s''|s, a) P(s''|s', a')}{4(q(s'') - q(s'', a''))^{3/2}}; \tag{11}$$

- *for any $k = 1, \ldots, L$, $s \in S_{k-1}$, and $a \in A$,*

$$\frac{\partial^2 \phi_H}{\partial q(s, a)^2} = \frac{1}{4q(s, a)^{3/2}} + \frac{\alpha}{4(q(s) - q(s, a))^{3/2}} + \sum_{s' \in S_k, s' \neq s_L} \sum_{a' \in A} \frac{\alpha P(s'|s, a)^2}{4(q(s') - q(s', a'))^{3/2}}; \tag{12}$$

- *all other entries of the Hessian are $0$.*

*Moreover, for any $w : (S \setminus \{s_L\}) \times A \to \mathbb{R}$, the quadratic form $w^\top \nabla^2 \phi_H(q) w$ can be written as*

$$\frac{1}{4} \sum_{s \neq s_L} \sum_a \left( \frac{w(s, a)^2}{q(s, a)^{3/2}} + \frac{\alpha(h(s) - w(s, a))^2}{(q(s) - q(s, a))^{3/2}} \right)$$
$$\text{where} \quad h(s) = \begin{cases} \sum_{s' \in S_{k(s)-1}} \sum_{a' \in A} P(s|s', a') w(s', a'), & \text{if } k(s) > 0, \\ 0, & \text{else.} \end{cases} \tag{13}$$

*Consequently, $\phi_H$ is convex in $q$.*

*Proof.* Fix a state-action pair $(s, a)$. By direct calculation, we show that the terms in $\phi_H(q)$ containing the variable $q(s, a)$ are

$$- \left( \sqrt{q(s, a)} + \alpha \sqrt{q(s) - q(s, a)} \right) - \alpha \sum_{s'' \in S_{k(s)+1}} \sum_{a'' \in A} \sqrt{q(s'') - q(s'', a'')} \tag{14}$$

where the last term is zero when $s = s_L$. From Eq. (14), we can infer that the second-order partial derivatives of $q(s, a)$ and $q(s', a')$ are non-zero if and only if $|k(s) - k(s')| = 1$.

We first verify Eq. (10), where $k(s') = k(s) - 1$ and the derivatives are from $-\alpha \sqrt{q(s) - q(s, a)}$. Direct calculations shows

$$\frac{\partial^2 \phi_H}{\partial q(s', a') \partial q(s, a)} = \frac{\partial^2}{\partial q(s', a') \partial q(s, a)} \left( -\alpha \sqrt{q(s) - q(s, a)} \right)$$
$$= \frac{\partial}{\partial q(s, a)} \left( \frac{-\alpha P(s|s', a')}{2\sqrt{q(s) - q(s, a)}} \right)$$
$$= \frac{-\alpha P(s|s', a')}{4 \left(q(s) - q(s, a)\right)^{3/2}}$$

where the second step follows because the term $q(s', a') P(s|s', a')$ belongs to $q(s)$ by Eq. (9).

For the second case (Eq. (11)) where $k(s) = k(s')$ and $(s, a) \neq (s', a')$, we have that

$$\frac{\partial^2 \phi_H}{\partial q(s', a') \partial q(s, a)} = \frac{\partial^2}{\partial q(s', a') \partial q(s, a)} \left( -\alpha \sum_{s'' \in S_{k(s)+1}} \sum_{a'' \in A} \sqrt{q(s'') - q(s'', a'')} \right)$$

$$= \sum_{s'' \in S_{k(s)+1}} \sum_{a'' \in A} \frac{\partial^2}{\partial q(s',a')\partial q(s,a)} \left( -\alpha\sqrt{q(s'') - q(s'',a'')} \right)$$

$$= \sum_{s'' \in S_{k(s)+1}} \sum_{a'' \in A} \frac{\partial}{\partial q(s,a)} \left( \frac{-\alpha P(s''|s',a')}{2\sqrt{q(s'') - q(s'',a'')}} \right)$$

$$= \sum_{s'' \in S_k} \sum_{a'' \in A} \frac{\alpha P(s''|s,a)P(s''|s',a')}{4(q(s'') - q(s'',a''))^{3/2}}$$

where the third step follows from the previous calculation.

Finally, we finish the first part of the proof by verifying Eq. (12) that

$$\frac{\partial^2 \phi_H}{\partial q(s,a)^2} = -\frac{\partial^2}{\partial q(s,a)^2} \left( \sqrt{q(s,a)} + \alpha\sqrt{q(s) - q(s,a)} \right)$$

$$- \sum_{s'' \in S_{k(s)+1}} \sum_{a'' \in A} \frac{\partial^2}{\partial q(s,a)^2} \left( \alpha\sqrt{q(s'') - q(s'',a'')} \right)$$

$$= \frac{1}{4q(s,a)^{3/2}} + \frac{\alpha}{4(q(s) - q(s,a))^{3/2}} + \sum_{s' \in S_k, s' \neq s_L} \sum_{a' \in A} \frac{\alpha P(s'|s,a)^2}{4(q(s') - q(s',a'))^{3/2}}.$$

Then, we are ready to prove Eq. (13). Indeed, we have

$$w^\top \nabla^2 \phi_H(q)w = \sum_{s \neq s_L} \sum_a \left( \frac{w(s,a)^2}{4q(s,a)^{3/2}} + \frac{\alpha w(s,a)^2}{4(q(s) - q(s,a))^{3/2}} \right)$$

$$+ \sum_{0 < k < L} \sum_{s \in S_{k-1}} \sum_a w(s,a)^2 \sum_{s'' \in S_k} \sum_{a'' \in A} \frac{\alpha P(s''|s,a)^2}{4(q(s'') - q(s'',a''))^{3/2}}$$

$$+ \sum_{0 < k < L} \sum_{s,s' \in S_{k-1}} \sum_{a,a' \in A:(s,a) \neq (s',a')} w(s,a)w(s',a') \sum_{s'' \in S_k} \sum_{a'' \in A} \frac{\alpha P(s''|s,a)P(s''|s',a')}{4(q(s'') - q(s'',a''))^{3/2}}$$

$$- 2\sum_{0 < k < L} \sum_{s \in S_k} \sum_{s' \in S_{k-1}} \sum_{a,a'} \frac{\alpha P(s|s',a')w(s,a)w(s',a')}{4(q(s) - q(s,a))^{3/2}}$$

$$= \sum_{s \neq s_L} \sum_a \left( \frac{w(s,a)^2}{4q(s,a)^{3/2}} + \frac{\alpha w(s,a)^2}{4(q(s) - q(s,a))^{3/2}} \right)$$

$$+ \sum_{0 < k < L} \sum_{s'' \in S_k} \sum_{a'' \in A} \frac{\alpha}{4(q(s'') - q(s'',a''))^{3/2}} \left( \sum_{s \in S_{k-1}} \sum_a w(s,a)P(s''|s,a) \right)^2$$

$$- 2\sum_{0 < k < L} \sum_{s \in S_k} \sum_a \frac{\alpha w(s,a) \sum_{s' \in S_{k-1}} \sum_{a'} P(s|s',a')w(s',a')}{4(q(s) - q(s,a))^{3/2}}$$

$$= \sum_{s \neq s_L} \sum_a \left( \frac{w(s,a)^2}{4q(s,a)^{3/2}} + \frac{\alpha w(s,a)^2}{4(q(s) - q(s,a))^{3/2}} \right)$$

$$+ \sum_{0 < k < L} \sum_{s \in S_k} \sum_{a \in A} \frac{\alpha h(s)^2}{4(q(s) - q(s,a))^{3/2}} - 2\sum_{0 < k < L} \sum_{s \in S_k} \sum_a \frac{\alpha w(s,a)h(s)}{4(q(s) - q(s,a))^{3/2}}$$

$$= \frac{1}{4} \sum_{s \neq s_L} \sum_a \left( \frac{w(s,a)^2}{q(s,a)^{3/2}} + \frac{\alpha(h(s) - w(s,a))^2}{(q(s) - q(s,a))^{3/2}} \right),$$

finishing the proof. $\qquad\square$

Note that the Hessian is clearly non-diagonal. However, by using the alternative definition from Eq. (9), the Hessian has a layered structure which allows an induction-based analysis as we will show.

# B Proof of Lemma 5

In this section, we provide the proof for Lemma 5. First, we introduce the following notations.

$$F_t(q) = \left\langle q, \widehat{L}_{t-1} \right\rangle + \psi_t(q) , \quad G_t = \left\langle q_t, \widehat{L}_t \right\rangle + \psi_t(q),$$

$$q_t = \underset{q \in \Omega}{\operatorname{argmin}} F_t(q) , \quad \widetilde{q}_t = \underset{q \in \Omega}{\operatorname{argmin}} G_t(q). \tag{15}$$

Note that the definition of $q_t$ is consistent with Algorithm 1. With these notations, we decompose the regret $\mathbb{E}\left[\sum_{t=1}^{T} \left\langle q_t - u, \widehat{\ell}_t \right\rangle\right]$ against any occupancy measure $u \in \Omega$ into a stability term and a mixed penalty term by adding and subtracting $F_t(q_t)$ and $G_t(\widetilde{q}_t)$:

$$= \underbrace{\mathbb{E}\left[\sum_{t=1}^{T} \left(\left\langle q_t, \widehat{\ell}_t \right\rangle + F_t(q_t) - G_t(\widetilde{q}_t)\right)\right]}_{\text{stability}} + \underbrace{\mathbb{E}\left[\sum_{t=1}^{T} \left(G_t(\widetilde{q}_t) - F_t(q_t) - \left\langle u, \widehat{\ell}_t \right\rangle\right)\right]}_{\text{mixed penalty}}. \tag{16}$$

The rest of the section is organized as follows:

1. With some auxiliary lemmas (Lemma 10 and Lemma 11), we prove in Lemma 12 that the update of the algorithm is smooth in the sense that $q_t$ and $\widetilde{q}_t$ are close.

2. With the smoothness guarantee, we bound the stability term by the quadratic norm of $\widehat{\ell}_t$ with respect to the Hessian matrix in Lemma 13, using similar techniques from [16].

3. By standard analysis, we control the mixed penalty term in Lemma 14.

4. Picking a proper $u$ that is closed to $\mathring{q}$ and specified in Lemma 15, we finish the proof of Lemma 5 in the end of this section.

We use the notation $M_1 \preceq M_2$ for two matrices $M_1$ and $M_2$ to denote the fact that $M_2 - M_1$ is positive semi-definite.

**Lemma 10** (Convexity of $\Omega$). *The set of valid occupancy measures $\Omega$ is convex.*

*Proof.* For any $u, v \in \Omega$ and $\lambda \in [0, 1]$, it suffices to verify that $p = \lambda u + (1 - \lambda)v$ satisfies the two constraints described in Section 2. For the first one: we have for any $k = 0, \ldots, L - 1$,

$$\sum_{s \in S_k} \sum_{a \in A} p(s, a) = \lambda \sum_{s \in S_k} \sum_{a \in A} u(s, a) + (1 - \lambda) \sum_{s \in S_k} \sum_{a \in A} v(s, a)$$

$$= \lambda + (1 - \lambda) = 1.$$

For the second one (Eq. (1)), we have for any $k = 1, \ldots, L - 1$ and every state $s \in S_k$,

$$\sum_{s' \in S_{k-1}, a'} P(s|s', a')p(s', a')$$

$$= \sum_{s' \in S_{k-1}, a'} P(s|s', a') \left(\lambda u(s', a') + (1 - \lambda)v(s', a')\right)$$

$$= \lambda \sum_{s' \in S_{k-1}, a'} P(s|s', a')u(s', a') + (1 - \lambda) \sum_{s' \in S_{k-1}, a'} P(s|s', a')v(s', a')$$

$$= \lambda \sum_{a} u(s, a) + (1 - \lambda) \sum_{a} v(s, a)$$

$$= \sum_{a} \left(\lambda u(s, a) + (1 - \lambda)v(s, a)\right)$$

$$= \sum_{a} p(s, a).$$

This finishes the proof. $\qquad\qquad\square$

**Lemma 11.** *For any occupancy measures $q$ and $p$ from $\Omega$ satisfying $\frac{1}{2}q(s,a) \leq p(s,a)$ for all state-action pair $(s,a)$, we have $\frac{1}{4}\nabla^2\psi_t(p) \preceq \nabla^2\psi_t(q)$.*

*Proof.* Let $M_1 = \nabla^2\psi_t(p)$ and $M_2 = \nabla^2\psi_t(q)$. By Eq. (13) in Lemma 9, we have for any $w$,

$$w^\top M_2 w = \frac{1}{4\eta_t} \sum_{s \neq s_L, a} \left\{ \frac{w(s,a)^2}{q(s,a)^{3/2}} + \frac{\alpha\,(h(s) - w(s,a))^2}{(q(s) - q(s,a))^{3/2}} \right\} + \beta \sum_{s \neq s_L, a} \frac{w(s,a)^2}{q(s,a)^2}.$$

According to the condition of the lemma and the fact $q(s) - q(s,a) = \sum_{b \neq a} q(s,b)$ and $p(s) - p(s,a) = \sum_{b \neq a} p(s,b)$, we have $\frac{1}{2}\,(q(s) - q(s,a)) \leq p(s) - p(s,a)$, and thus

$$w^\top M_2 w \geq \frac{1}{4\eta_t} \sum_{s \neq s_L, a} \left\{ \frac{w(s,a)^2}{(2p(s,a))^{3/2}} + \frac{\alpha\,(h(s) - w(s,a))^2}{(2(p(s) - p(s,a)))^{3/2}} \right\} + \beta \sum_{s,a} \frac{w(s,a)^2}{(2p(s,a))^2}$$

$$\geq \frac{1}{16\eta_t} \sum_{s \neq s_L, a} \left\{ \frac{w(s,a)^2}{p(s,a)^{3/2}} + \frac{\alpha\,(h(s) - w(s,a))^2}{(p(s) - p(s,a))^{3/2}} \right\} + \frac{\beta}{4} \sum_{s,a} \frac{w(s,a)^2}{p(s,a)^2}$$

$$= \frac{1}{4} w^\top M_1 w.$$

This finishes the proof. $\qquad\square$

With the help of Lemma 10 and Lemma 11, we now prove that $q_t$ to $\widetilde{q}_t$ are close.

**Lemma 12.** *With $\beta = 64L$, we have $\frac{1}{2}q_t(s,a) \leq \tilde{q}_t(s,a) \leq 2q_t(s,a)$ for all state-action pairs $(s,a)$ (recall the definitions in Eq. (15)).*

*Proof.* For simplicity, we denote $H$ as the Hessian $\nabla^2\psi_t(q_t)$, and $H_L$ as the Hessian $\nabla^2\phi_L(q_t)$ which is a diagonal matrix with $\frac{\beta}{q_t(s,a)^2}$ on the diagonal. Recalling that $\psi_t = \eta_t^{-1}\phi_H + \phi_L$, by the convexity of $\phi_H$ (Lemma 9), we have $H_L \preceq H$. Our goal is to prove $\|\tilde{q}_t - q_t\|_H \leq 1$. This is enough to prove the lemma statement because

$$1 \geq \|\tilde{q}_t - q_t\|_H \geq \|\tilde{q}_t - q_t\|_{H_L} = \beta \sum_{s,a} \frac{(\tilde{q}_t(s,a) - q_t(s,a))^2}{q_t(s,a)^2}, \tag{17}$$

and since $\beta \geq 9$, we have $(\tilde{q}_t(s,a) - q_t(s,a))^2 \leq \frac{q_t(s,a)^2}{\beta} \leq \left(\frac{q_t(s,a)}{3}\right)^2$, which indicates $\frac{1}{2}q_t(s,a) \leq \tilde{q}_t(s,a) \leq 2q_t(s,a)$.

To prove $\|\tilde{q}_t - q_t\|_H \leq 1$, it suffices to show that for any occupancy measure $q'$ that satisfies $\|q' - q_t\|_H = 1$, we have $G_t(q') \geq G_t(q_t)$, because this implies that, as the minimizer of the convex function $G_t$, $\tilde{q}_t$ must be within the convex set $\{q : \|q - q_t\|_H \leq 1\}$. To this end, we bound $G_t(q')$ as follows

$$G_t(q') = G_t(q_t) + \nabla G_t(q_t)^\top (q' - q_t) + \frac{1}{2} \|q' - q_t\|_{\nabla^2\psi_t(\xi)}^2$$

$$= G_t(q_t) + \nabla F_t(q_t)^\top (q' - q_t) + \widehat{\ell}_t^\top (q' - q_t) + \frac{1}{2} \|q' - q_t\|_{\nabla^2\psi_t(\xi)}^2$$

$$\geq G_t(q_t) - \left\|\widehat{\ell}_t\right\|_{H^{-1}} \|q' - q_t\|_H + \frac{1}{2} \|q' - q_t\|_{\nabla^2\psi_t(\xi)}^2$$

$$= G_t(q_t) - \left\|\widehat{\ell}_t\right\|_{H^{-1}} + \frac{1}{2} \|q' - q_t\|_{\nabla^2\psi_t(\xi)}^2$$

where in the first step we use Taylor's expansion with $\xi$ being a point between $q_t$ and $q'$, in the second step we use the definition of $G_t$ and $F_t$ (in Eq. (15)), and in the third step we use Hölder's inequality and the first order optimality condition $\langle \nabla F_t(q_t), q' - q_t \rangle \geq 0$.

Repeating the earlier argument in Eq. (17), $\|q' - q_t\|_H = 1$ also implies $\frac{1}{2}q_t(s,a) \leq q'(s,a) \leq 2q_t(s,a)$ and thus $\frac{1}{2}q_t(s,a) \leq \xi(s,a) \leq 2q_t(s,a)$. Since $\xi$ is in $\Omega$ by Lemma 10, we continue to bound the last expression using Lemma 11:

$$G_t(q_t) - \left\|\widehat{\ell}_t\right\|_{H^{-1}} + \frac{1}{2} \|q' - q_t\|_{\nabla^2\psi_t(\xi)}^2$$

$$\geq G_t(q_t) - \left\|\widehat{\ell}_t\right\|_{H^{-1}} + \tfrac{1}{8} \|q' - q\|_H^2$$

$$= G_t(q_t) - \left\|\widehat{\ell}_t\right\|_{H^{-1}} + \tfrac{1}{8}. \tag{18}$$

Finally, we bound the term $\left\|\widehat{\ell}_t\right\|_{H^{-1}}$. By the definition of $\widehat{\ell}_t$, with shorthand $\mathbb{I}(s,a) \triangleq \mathbb{I}\{s_{k(s)} = s, a_{k(s)} = a\}$ we show that

$$\left\|\widehat{\ell}_t\right\|_{H^{-1}}^2 \leq \left\|\widehat{\ell}_t\right\|_{H_L^{-1}}^2 \qquad\qquad (H_L \preceq H)$$

$$= \sum_{s,a} \frac{\mathbb{I}(s,a)\ell_t(s,a)^2}{q_t(s,a)^2} \frac{q_t(s,a)^2}{\beta}$$

$$\leq \sum_{s,a} \frac{\mathbb{I}(s,a)}{\beta} = \frac{L}{\beta} = \frac{1}{64}.$$

Plugging it into Eq. (18), we conclude that $G_t(q') \geq G_t(q_t)$, which finishes the proof. $\qquad\square$

We are now ready to bound the first term in Eq. (16).

**Lemma 13.** *With $\beta = 64L$, we have*

$$\sum_{t=1}^{T} \left( \left\langle q_t, \widehat{\ell}_t \right\rangle + F_t(q_t) - G_t(\tilde{q}_t) \right) \leq 8 \sum_{t=1}^{T} \left\|\widehat{\ell}_t\right\|_{\nabla^{-2}\psi_t(q_t)}^2.$$

*Proof.* We first lower bound the term $\left\langle q_t, \widehat{\ell}_t \right\rangle + F_t(q_t) - G_t(\tilde{q}_t)$ as

$$\left\langle q_t, \widehat{\ell}_t \right\rangle + F_t(q_t) - G_t(\tilde{q}_t)$$

$$= \left\langle q_t, \widehat{\ell}_t + \widehat{L}_{t-1} \right\rangle + \psi_t(q_t) - G_t(\tilde{q}_t)$$

$$= G_t(q_t) - G_t(\tilde{q}_t)$$

$$= \langle \nabla G_t(\tilde{q}_t), q_t - \tilde{q}_t \rangle + \tfrac{1}{2} \|q_t - \tilde{q}_t\|_{\nabla^2\psi_t(\xi)}^2$$

$$\geq \tfrac{1}{2} \|q_t - \tilde{q}_t\|_{\nabla^2\psi_t(\xi)}^2,$$

where in the second to last step we apply Taylor's expansion with $\xi$ being a point between $q_t$ and $\tilde{q}_t$, and in the last step we use the first order optimality condition of $\tilde{q}_t$.

On the other hand, we can upper bound this term as

$$\left\langle q_t, \widehat{\ell}_t \right\rangle + F_t(q_t) - G_t(\tilde{q}_t)$$

$$= \left\langle q_t - \tilde{q}_t, \widehat{\ell}_t \right\rangle + F_t(q_t) - F_t(\tilde{q}_t)$$

$$\leq \left\langle q_t - \tilde{q}_t, \widehat{\ell}_t \right\rangle$$

$$\leq \|q_t - \tilde{q}_t\|_{\nabla^2\psi_t(\xi)} \left\|\widehat{\ell}_t\right\|_{\nabla^{-2}\psi_t(\xi)},$$

where the second step is by the optimality of $q_t$ and the last step is by Hölder's inequality. Combining the lower and upper bounds we arrive at:

$$\left\langle q_t, \widehat{\ell}_t \right\rangle + F_t(q_t) - G_t(\tilde{q}_t) \leq 2 \left\|\widehat{\ell}_t\right\|_{\nabla^{-2}\psi_t(\xi)}^2.$$

Moreover, by Lemma 12, we know that $\tilde{q}_t$ satisfies $\frac{1}{2}q_t(s,a) \leq \tilde{q}_t(s,a) \leq 2q_t(s,a)$ for all state-action pairs. Since $\xi$ is a middle point between $q_t$ and $\tilde{q}_t$, it satisfies $\frac{1}{2}q_t(s,a) \leq \xi(s,a) \leq 2q_t(s,a)$ for all state-action pairs as well. According to Lemma 11, we then have $\left\|\widehat{\ell}_t\right\|_{\nabla^{-2}\psi_t(\xi)}^2 \leq 4 \left\|\widehat{\ell}_t\right\|_{\nabla^{-2}\psi_t(q_t)}^2$, which completes the proof. $\qquad\square$

Next, we bound the second term in Eq. (16).

**Lemma 14.** *For any $u \in \Omega$, we have (with $1/\eta_0 \triangleq 0$)*

$$\sum_{t=1}^{T} \left( G_t(\tilde{q}_t) - F_t(q_t) - \left\langle u, \widehat{\ell}_t \right\rangle \right) = \phi_L(u) - \phi_L(q_1) + \sum_{t=1}^{T} \left( \frac{1}{\eta_t} - \frac{1}{\eta_{t-1}} \right) (\phi_H(u) - \phi_H(q_t)).$$

*Proof.* Due to the optimality of $\tilde{q}_t$, we have $G_t(\tilde{q}_t) \leq G_t(q_{t+1})$ and also $G_T(\tilde{q}_T) \leq G_T(u)$. With the help of these inequalities, we proceed as

$$\sum_{t=1}^{T} \left( G_t(\tilde{q}_t) - F_t(q_t) - \left\langle u, \widehat{\ell}_t \right\rangle \right)$$

$$\leq \sum_{t=1}^{T} \left( G_t(q_{t+1}) - F_t(q_t) - \left\langle u, \widehat{\ell}_t \right\rangle \right)$$

$$= -F_1(q_1) + \sum_{t=2}^{T} (G_{t-1}(q_t) - F_t(q_t)) + G_T(u) - \left\langle u, \widehat{L}_T \right\rangle$$

$$= -\psi_1(q_1) - \sum_{t=2}^{T} \left( \frac{1}{\eta_t} - \frac{1}{\eta_{t-1}} \right) \phi_H(q_t) + \psi_T(u)$$

$$= -\phi_H(q_1) - \phi_L(q_1) - \sum_{t=2}^{T} \left( \frac{1}{\eta_t} - \frac{1}{\eta_{t-1}} \right) \phi_H(q_t) + \frac{\phi_H(u)}{\eta_T} + \phi_L(u)$$

$$= \phi_L(u) - \phi_L(q_1) + \sum_{t=1}^{T} \left( \frac{1}{\eta_t} - \frac{1}{\eta_{t-1}} \right) (\phi_H(u) - \phi_H(q_t)),$$

finishing the proof. $\qquad\square$

While it is tempting to set $u = \mathring{q}$ to obtain a regret bound, note that $\phi_L(\mathring{q})$ can potentially grow to infinity. To this end, we will set $u$ as some point close enough to $\mathring{q}$, specified in the following lemma.

**Lemma 15.** *Suppose $\beta = 64L$. Let $v = \left(1 - \frac{1}{T}\right)\mathring{q} + \frac{q_1}{T}$, where $\mathring{q}$ is the optimal occupancy measure and $q_1 = \operatorname{argmin}_{q \in \Omega} \psi_1(q)$ is the initial occupancy measure. Then $v$ satisfies the following:*

- $\mathbb{E}\left[ \sum_{t=1}^{T} \left\langle v - \mathring{q}, \widehat{\ell}_t \right\rangle \right] \leq 2L$,
- $\phi_L(v) - \phi_L(q_1) \leq 64L|S||A| \log T$,
- $\phi_H(v) - \phi_H(\mathring{q}) \leq \frac{(1+\alpha)|S||A|}{T}$.

*Proof.* The first statement is by direct calculation:

$$\mathbb{E}\left[ \sum_{t=1}^{T} \left\langle v - \mathring{q}, \widehat{\ell}_t \right\rangle \right] = \frac{1}{T} \mathbb{E}\left[ \sum_{t=1}^{T} \left\langle q_1 - \mathring{q}, \widehat{\ell}_t \right\rangle \right]$$

$$= \frac{1}{T} \left\langle q_1 - \mathring{q}, \sum_{t=1}^{T} \ell_t \right\rangle$$

$$\leq \frac{1}{T} \|q_1 - \mathring{q}\|_1 \left\| \sum_{t=1}^{T} \ell_t \right\|_\infty \leq \frac{2LT}{T} = 2L.$$

The second statement directly uses the definition of $\phi_L$:

$$\phi_L(v) - \phi_L(q_1) = 64L \sum_{s,a} \log\left( \frac{q_1(s,a)}{v(s,a)} \right) \leq 64L|S||A| \log(T).$$

Finally, we verify the last statement:

$$
\phi_H(v) - \phi_H(\mathring{q})
$$

$$
= \sum_{s,a} \left( \sqrt{\mathring{q}(s,a)} + \alpha \sqrt{\mathring{q}(s) - \mathring{q}(s,a)} - \sqrt{v(s,a)} - \alpha \sqrt{v(s) - v(s,a)} \right)
$$

$$
= \sum_{s,a} \left( \sqrt{\mathring{q}(s,a)} - \sqrt{\frac{T-1}{T} \mathring{q}(s,a) + \frac{q_1(s,a)}{T}} \right)
$$

$$
+ \sum_{s,a} \alpha \left( \sqrt{\mathring{q}(s) - \mathring{q}(s,a)} - \sqrt{\frac{T-1}{T}(\mathring{q}(s) - \mathring{q}(s,a)) + \frac{q_1(s) - q_1(s,a)}{T}} \right)
$$

$$
\le \left( 1 - \sqrt{\frac{T-1}{T}} \right) \sum_{s,a} \left( \sqrt{\mathring{q}(s,a)} + \alpha \sqrt{\mathring{q}(s) - \mathring{q}(s,a)} \right)
$$

$$
= \frac{\left( \sqrt{T} - \sqrt{T-1} \right) \left( \sqrt{T} + \sqrt{T-1} \right)}{\sqrt{T} \left( \sqrt{T} + \sqrt{T-1} \right)} \left( (1+\alpha)|S||A| \right)
$$

$$
\le \frac{(1+\alpha)|S||A|}{T}.
$$

$\square$

Finally, we are ready to prove Lemma 5.

*Proof of Lemma 5.* By combining the Lemma 14 and 13 and taking expectation on both sides, we have that $\mathbb{E}\left[ \sum_{t=1}^{T} \left\langle q_t - u, \widehat{\ell}_t \right\rangle \right]$ is bounded by

$$
\phi_L(u) - \phi_L(q_1) + \sum_{t=1}^{T} \left( \frac{1}{\eta_t} - \frac{1}{\eta_{t-1}} \right) \mathbb{E}\left[ \phi_H(u) - \phi_H(q_t) \right] + 8 \sum_{t=1}^{T} \mathbb{E}\left[ \left\| \widehat{\ell}_t \right\|_{\nabla^{-2}\psi_t(q_t)}^2 \right].
$$

Picking the intermediate occupancy measure $v = \left( 1 - \frac{1}{T} \right) \mathring{q} + \frac{1}{T} q_1$, by Lemma 15, we have

$$
\mathrm{Reg}_T = \mathbb{E}\left[ \sum_{t=1}^{T} \left\langle q_t - \mathring{q}, \widehat{\ell}_t \right\rangle \right]
$$

$$
= \mathbb{E}\left[ \sum_{t=1}^{T} \left\langle q_t - v, \widehat{\ell}_t \right\rangle \right] + \mathbb{E}\left[ \sum_{t=1}^{T} \left\langle v - \mathring{q}, \widehat{\ell}_t \right\rangle \right]
$$

$$
\le 2L + 64L|S||A|\log T + \sum_{t=1}^{T} \left( \frac{1}{\eta_t} - \frac{1}{\eta_{t-1}} \right) \mathbb{E}\left[ \phi_H(v) - \phi_H(q_t) \right] + 8 \sum_{t=1}^{T} \mathbb{E}\left[ \left\| \widehat{\ell}_t \right\|_{\nabla^{-2}\psi_t(q_t)}^2 \right]
$$

$$
\le 2L + 64L|S||A|\log T + 8 \sum_{t=1}^{T} \mathbb{E}\left[ \left\| \widehat{\ell}_t \right\|_{\nabla^{-2}\psi_t(q_t)}^2 \right]
$$

$$
+ \left( \frac{1}{\eta_T} - \frac{1}{\eta_0} \right) \mathbb{E}\left[ \phi_H(v) - \phi_H(\mathring{q}) \right] + \sum_{t=1}^{T} \left( \frac{1}{\eta_t} - \frac{1}{\eta_{t-1}} \right) \mathbb{E}\left[ \phi_H(\mathring{q}) - \phi_H(q_t) \right]
$$

$$
\le \mathcal{O}\left( L|S||A|\log T \right) + \sum_{t=1}^{T} \left( \frac{1}{\eta_t} - \frac{1}{\eta_{t-1}} \right) \mathbb{E}\left[ \phi_H(\mathring{q}) - \phi_H(q_t) \right] + 8 \sum_{t=1}^{T} \mathbb{E}\left[ \left\| \widehat{\ell}_t \right\|_{\nabla^{-2}\psi_t(q_t)}^2 \right]
$$

where the last line follows from the fact $\frac{(1+\alpha)|S||A|}{\gamma\sqrt{T}} = \mathcal{O}(1)$. $\square$

## C  Proof of Lemma 6

Recall the notation defined in Section 4:

$$h_s(\pi) = \sum_{a \in A} \sqrt{\pi(a|s)} + \alpha\sqrt{1 - \pi(a|s)}.$$

Clearly, for any mapping (deterministic policy) $\pi : S \to A$, we have $h_s(\pi) = 1 + \alpha(|A| - 1)$ for all state $s$. Therefore, we can decompose $\phi_H(\mathring{q}) - \phi_H(q_t)$ as

$$\phi_H(\mathring{q}) - \phi_H(q_t) = \sum_{s \neq s_L} \sqrt{q_t(s)} h_s(\pi_t) - \sum_{s \neq s_L} \sqrt{\mathring{q}(s)} h_s(\mathring{\pi})$$

$$= \sum_{s \neq s_L} \sqrt{q_t(s)} \left(h_s(\pi_t) - h_s(\mathring{\pi})\right) + (1 + \alpha(|A| - 1)) \sum_{s \neq s_L} \left(\sqrt{q_t(s)} - \sqrt{\mathring{q}(s)}\right).$$

In the rest of this section, we first bound the term $\sum_{s \neq s_L} \sqrt{q_t(s)} \left(h_s(\pi_t) - h_s(\mathring{\pi})\right)$ in Lemma 16, and then show that the term $\sum_{s \neq s_L} \left(\sqrt{q_t(s)} - \sqrt{\mathring{q}(s)}\right)$ can be bounded as

$$\sum_{s \neq s_L} \left(\sqrt{q_t(s)} - \sqrt{\mathring{q}(s)}\right) = \sum_{s \neq s_L} \left(\sqrt{q_t(s)} - \sqrt{q^\pi(s)}\right) + \sum_{s \neq s_L} \left(\sqrt{q^\pi(s)} - \sqrt{\mathring{q}(s)}\right)$$

$$\leq \sqrt{|S|L} \left(\sqrt{\sum_{s \neq s_L} \sum_{a \neq \pi(s)} q_t(s,a)} + \sqrt{\sum_{s \neq s_L} \sum_{a \neq \pi(s)} \mathring{q}(s,a)}\right)$$

$$\leq 2\sqrt{|S|L \sum_{s \neq s_L} \sum_{a \neq \pi(s)} q_t(s,a) + \mathring{q}(s,a)}$$

for any mapping $\pi$, where the second line follows Lemma 19 and Lemma 20, in which we apply the key induction Lemma 18 based on the *state reach probability* defined in Lemma 17.

**Lemma 16.** *For any policy $\pi_1$ and mapping $\pi_2 : S \to A$, we have*

$$\sum_{s \neq s_L} \sqrt{q_1(s)} \left(h_s(\pi_1) - h_s(\pi_2)\right) \leq (1 + \alpha) \sum_{s \neq s_L} \sum_{a \neq \pi(s)} \sqrt{q_1(s,a)},$$

*where $\pi$ is any mapping from $S$ to $A$ and $q_1 = q^{\pi_1}$.*

*Proof.* Direct calculation shows:

$$h_s(\pi_1) - h_s(\pi_2)$$
$$= h_s(\pi_1) - 1 - \alpha(|A| - 1)$$
$$= \sqrt{\pi_1(\pi(s)|s)} + \alpha\sqrt{1 - \pi_1(\pi(s)|s)} - 1 + \sum_{a \neq \pi(s)} \left(\sqrt{\pi_1(a|s)} + \alpha\sqrt{1 - \pi_1(a|s)} - \alpha\right)$$
$$\leq \alpha\sqrt{1 - \pi_1(\pi(s)|s)} + \sum_{a \neq \pi(s)} \sqrt{\pi_1(a|s)}$$
$$= \alpha\sqrt{\sum_{a \neq \pi(s)} \pi_1(a|s)} + \sum_{a \neq \pi(s)} \sqrt{\pi_1(a|s)}$$
$$\leq (\alpha + 1) \sum_{a \neq \pi(s)} \sqrt{\pi_1(a|s)}.$$

Multiplying both sides by $\sqrt{q_1(s)}$ and summing over $s$ prove the lemma. $\square$

**Lemma 17.** *For any policy $\pi$, define its associated reachability probability $p^\pi : S \times A \times S \to [0,1]$ as*

$$p^\pi(s'|s,a) = \begin{cases} 0, & \text{if } k(s') \leq k(s), \\ P(s'|s,a), & \text{if } k(s') = k(s) + 1, \quad (19) \\ \sum_{s_m \in S_{k(s')-1}} p^\pi(s_m|s,a) \sum_a \pi(a|s_m) P(s'|s_m,a), & \text{if } k(s') > k(s) + 1, \end{cases}$$

*which is simply the probability of reaching state $s'$ after taking action $a$ at state $s$ and then following policy $\pi$. For any state-action pair $(s, a)$, policy $\pi$, and $k = 0, \ldots, L - 1$, we have*

$$\sum_{s' \in S_k} p^\pi(s'|s, a) \leq 1, \tag{20}$$

*which implies $\sum_{s \neq s_L} p^\pi(s'|s, a) \leq L$.*

*Proof.* Eq. (20) is clear just based on the definition of $p^\pi$. We provide a proof by induction for completeness. Clearly, it holds for all layer $l$ with $l \leq k(s)$ and also $l = k(s) + 1$ where $\sum_{s' \in S_l} p^\pi(s'|s, a) = \sum_{s' \in S_l} P(s'|s, a) = 1$. Now assume that Eq. (20) holds for some layer $l \geq k(s) + 1$. For layer $l + 1$, we have

$$\sum_{s' \in S_{l+1}} p^\pi(s'|s, a)$$

$$= \sum_{s' \in S_{l+1}} \left( \sum_{s_m \in S_l} p^\pi(s_m|s, a) \sum_a \pi(a|s_m) P(s'|s_m, a) \right)$$

$$= \sum_{s_m \in S_l} p^\pi(s_m|s, a) \left( \sum_{s' \in S_{l+1}} \sum_a \pi(a|s_m) P(s'|s_m, a) \right)$$

$$= \sum_{s_m \in S_l} p^\pi(s_m|s, a) \leq 1,$$

finishing the proof. □

Based on the concept of reachability probability, we prove the following key induction lemma.

**Lemma 18.** *If a policy $\pi$ and non-negative functions $f : S \times A \to \mathbb{R}_+ \cup \{0\}$ and $g : S \times A \to \mathbb{R}_+ \cup \{0\}$ satisfy*

$$f(s) \leq \sum_{s' \in S_{k(s)-1}} \sum_{a' \in A} g(s', a') P(s|s', a') + \sum_{s' \in S_{k(s)-1}} f(s') \left\{ \sum_{a' \in A} \pi(a'|s') P(s|s', a') \right\}, \forall s \neq s_0$$

*and $f(s_0) = 0$, then we have for all $s \neq s_0$,*

$$f(s) \leq \sum_{k=0}^{k(s)-1} \sum_{s' \in S_k} \sum_{a'} g(s', a') p^\pi(s|s', a'), \tag{21}$$

*where $p^\pi$ is the reachability probability defined in Eq. (19).*

*Proof.* We prove the statement by induction. First, for $s \in S_1$, using the condition of the lemma we have

$$f(s) \leq \sum_{s' \in S_0} \sum_{a'} g(s', a') P(s|s', a') + \sum_{s' \in S_0} f(s') \left\{ \sum_{a'} \pi(a'|s') P(s|s', a') \right\}$$

$$= \sum_{s' \in S_0} \sum_{a'} g(s', a') p^\pi(s|s', a')$$

where the second line follows from the fact $f(s_0) = 0$ and $p^\pi(s|s', a') = P(s|s', a')$ for $k(s) = k(s') + 1$. This proves the base case.

Now assume that Eq. (21) holds for all states in layer $l > 0$. For $s \in S_{l+1}$, we have

$$
\begin{aligned}
f(s) &\leq \sum_{s' \in S_l} \sum_{a'} g(s', a') P(s|s', a') + \sum_{s' \in S_l} f(s') \left\{ \sum_{a'} \pi(a'|s') P(s|s', a') \right\} \\
&\leq \sum_{s' \in S_l} \sum_{a'} g(s', a') P(s|s', a') \\
&\quad + \sum_{s' \in S_l} \left( \sum_{k=0}^{l-1} \sum_{s'' \in S_k} \sum_{a''} g(s'', a'') p^\pi(s'|s'', a'') \right) \left( \sum_{a'} \pi(a'|s') P(s|s', a') \right) \\
&= \sum_{s' \in S_l} \sum_{a'} g(s', a') p^\pi(s|s', a') \\
&\quad + \sum_{k=0}^{l-1} \sum_{s'' \in S_k} \sum_{a''} g(s'', a'') \left( \sum_{s' \in S_l} p^\pi(s'|s'', a'') \sum_{a'} \pi(a'|s') P(s|s', a') \right) \\
&= \sum_{s' \in S_l} \sum_{a'} g(s', a') p^\pi(s|s', a') + \sum_{k=0}^{l-1} \sum_{s'' \in S_k} \sum_{a''} g(s'', a'') p^\pi(s|s'', a'') \\
&= \sum_{k=0}^{l} \sum_{s' \in S_k} \sum_{a'} g(s', a') p^\pi(s|s', a'),
\end{aligned}
$$

where the second step uses the induction hypothesis and the fourth step uses the definition of the reachability probability. This finishes the induction. $\qquad\square$

We now apply the induction lemma to prove the following two key lemmas.

**Lemma 19.** *For any policy $\pi_1$ and mapping $\pi_2 : S \to A$, we have*

$$
\sum_{s \neq s_L} \sqrt{q_1(s)} - \sqrt{q_2(s)} \leq \sqrt{|S|L} \sqrt{\sum_{s \neq s_L} \sum_{a \neq \pi_2(s)} q_1(s, a)} \tag{22}
$$

*where we denote by $q_1 = q_1^\pi$ and $q_2 = q_2^\pi$ the occupancy measures of $\pi_1$ and $\pi_2$ respectively.*

*Proof.* We first bound $\sqrt{q_1(s)} - \sqrt{q_2(s)}$ by $\sqrt{\mathbb{I}_s (q_1(s) - q_2(s))}$ where $\mathbb{I}_s \triangleq \mathbb{I}\{q_1(s) \geq q_2(s)\}$. Define $f(s) = \mathbb{I}_s (q_1(s) - q_2(s))$ and $g(s, a) = \mathbb{I}\{a \neq \pi_2(s)\} q_1(s, a)$. Our goal is to prove for any $s \neq s_0$:

$$
f(s) \leq \sum_{s' \in S_{k(s)-1}} \sum_{a' \in A} g(s', a') P(s|s', a') + \sum_{s' \in S_{k(s)-1}} f(s') \left( \sum_{a' \in A} \pi_1(a'|s') P(s|s', a') \right) \tag{23}
$$

so that we can apply Lemma 18 (clearly we have $f(s_0) = 0$). To prove Eq. (23), consider a fixed state $s \in S_k$ for some $k > 0$. We rewrite $q_1(s) - q_2(s)$ as

$$
q_1(s) - q_2(s) = \sum_{s' \in S_{k-1}} \left[ q_1(s') \sum_{a' \in A} \pi_1(a'|s') P(s|s', a') - q_2(s') \sum_{a' \in A} \pi_2(s', a') P(s|s', a') \right].
$$

For state $s' \in S_{k-1}$ satisfying $q_1(s') \le q_2(s')$, we have

$$q_1(s') \sum_{a' \in A} \pi_1(a'|s')P(s|s',a') - q_2(s') \sum_{a' \in A} \pi_2(a'|s')P(s|s',a')$$

$$\le q_1(s') \sum_{a' \in A} \pi_1(a'|s')P(s|s',a') - q_1(s') \sum_{a' \in A} \pi_2(a'|s')P(s|s',a')$$

$$= q_1(s') \left( \sum_{a' \in A} (\pi_1(a'|s') - \pi_2(a'|s'))P(s|s',a') \right)$$

$$\le q_1(s') \left( \sum_{a' \ne \pi_2(s')} (\pi_1(a'|s'))P(s|s',a') \right)$$

$$= \sum_{a \ne \pi_2(s')} q_1(s',a')P(s|s',a').$$

where the forth line follows from the fact $\pi_1(a'|s') - \pi_2(a'|s') = \pi_1(a'|s') - 1 \le 0$ when $a' = \pi_2(s')$. For the other states with $q_1(s') \ge q_2(s')$, we have

$$q_1(s') \sum_{a' \in A} \pi_1(a'|s')P(s|s',a') - q_2(s') \sum_{a' \in A} \pi_2(a'|s')P(s|s',a')$$

$$= (q_1(s') - q_2(s')) \sum_{a' \in A} \pi_1(a'|s')P(s|s',a') + q_2(s') \sum_{a' \in A} (\pi_1(a'|s') - \pi_2(a'|s'))P(s'|s',a)$$

$$\le (q_1(s') - q_2(s')) \sum_{a' \in A} \pi_1(a'|s')P(s|s',a') + q_2(s') \sum_{a' \ne \pi_2(s')} \pi_1(a'|s')P(s'|s',a)$$

$$\le \mathbb{I}_{s'}(q_1(s') - q_2(s')) \sum_{a' \in A} \pi_1(a'|s')P(s|s',a') + \sum_{a' \ne \pi_2(s')} q_1(s',a')P(s'|s',a)$$

where the last line is due to the condition $q_1(s') \ge q_2(s')$.

Combining these two cases together yields Eq. (23). Therefore, applying Lemma 18 gives

$$\mathbb{I}_s(q_1(s) - q_2(s)) \le \sum_{k=0}^{k(s)-1} \sum_{s' \in S_k} \sum_{a'} g(s',a')p^{\pi_1}(s|s',a')$$

$$= \sum_{k=0}^{k(s)-1} \sum_{s' \in S_k} \sum_{a' \ne \pi_2(s')} q_1(s',a')p^{\pi_1}(s|s',a'),$$

for all $s \ne s_0$. Taking square root and summation over all states, we have

$$\sum_{s \ne s_L} \sqrt{\mathbb{I}_s(q_1(s) - q_2(s))} \le \sum_{s \ne s_L} \sqrt{\sum_{k=0}^{k(s)-1} \sum_{s' \in S_k} \sum_{a' \ne \pi_2(s')} q_1(s',a')p^{\pi_1}(s|s',a')}.$$

Notice that

$$\sum_{s \ne s_L} \sum_{k=0}^{k(s)-1} \sum_{s' \in S_k} \sum_{a' \ne \pi_2(s')} q_1(s',a')p^{\pi_1}(s|s',a') \le \sum_{s' \ne s_L} \sum_{a' \ne \pi_2(s')} q_1(s',a') \left( \sum_{s \ne s_L} p^{\pi_1}(s|s',a') \right)$$

$$\le L \sum_{s \ne s_L} \sum_{a \ne \pi_2(s)} q_1(s,a),$$

where the last step uses Lemma 17. Finally by Hölder's inequality, we arrive at

$$\sum_{s\neq s_L}\sqrt{q_1(s)}-\sqrt{q_2(s)}\leq\sum_{s\neq s_L}\sqrt{\sum_{k=0}^{k(s)-1}\sum_{s'\in S_k}\sum_{a'\neq\pi_2(s')}q_1(s',a')p^{\pi_1}(s|s',a')}$$

$$\leq\left(L\sum_{s\neq s_L}\sum_{a\neq\pi_2(s)}q_1(s,a)\right)^{1/2}\left(\sum_{s\neq s_L}1\right)^{1/2}$$

$$=\sqrt{|S|L}\sqrt{\sum_{s\neq s_L}\sum_{a\neq\pi_2(s)}q_1(s,a)},$$

which concludes the proof. □

**Lemma 20.** *For two deterministic policies $\pi_1$ and $\pi_2$, we have*

$$\sum_{s\neq s_L}\sqrt{q_1(s)}-\sqrt{q_2(s)}\leq\sqrt{|S|L}\sqrt{\sum_{s\neq s_L}\sum_{a\neq\pi_1(s)}q_2(s,a)}$$

*where we denote by $q_1=q^{\pi_1}$ and $q_2=q^{\pi_2}$ the occupancy measures of $\pi_1$ and $\pi_2$ respectively.*

*Proof.* The proof is similar to that of Lemma 19. First note that $\sqrt{q_1(s)}-\sqrt{q_2(s)}\leq\sqrt{\mathbb{I}_s(q_1(s)-q_2(s))}$ where $\mathbb{I}_s\triangleq\mathbb{I}\{q_1(s)\geq q_2(s)\}$. Define $f(s)=\mathbb{I}_s(q_1(s)-q_2(s))$ and $g(s,a)=\pi_1(a|s)\sum_{a'\neq\pi_1(s)}q_2(s,a')$. Our goal is to prove the following inequality

$$f(s)\leq\sum_{s'\in S_{k(s)-1}}\sum_{a'\in A}g(s',a')P(s|s',a')+\sum_{s'\in S_{k(s)-1}}f(s')\left(\sum_{a'\in A}\pi_1(a'|s')P(s|s',a')\right)$$

for all $s\neq s_L$ so that we can apply Lemma 18 again ($f(s_0)=0$ holds clearly). To show this, consider a fixed state $s\in S_k$ for some $k>0$ and rewrite the term $q_1(s)-q_2(s)$ as

$$q_1(s)-q_2(s)=\sum_{s'\in S_{k-1}}\sum_{a'}(q_1(s',a')P(s|s',a')-q_2(s',a')P(s|s',a'))$$

$$=\sum_{s'\in S_{k-1}}(q_1(s')P(s|s',\pi_1(s'))-q_2(s')P(s|s',\pi_2(s')))$$

since both $\pi_1$ and $\pi_2$ are deterministic polices. Then for the states $s'\in S_{k-1}$ with $\pi_1(s')=\pi_2(s')$, we have

$$q_1(s')P(s|s',\pi_1(s'))-q_2(s')P(s|s',\pi_2(s'))\leq\mathbb{I}_{s'}(q_1(s')-q_2(s'))P(s|s',\pi_1(s'))$$

For the other states with $\pi_1(s')\neq\pi_2(s')$, we have

$$q_1(s')P(s|s',\pi_1(s'))-q_2(s')P(s|s',\pi_2(s'))$$

$$\leq(q_1(s')-q_2(s'))P(s|s',\pi_1(s'))+q_2(s')P(s|s',\pi_1(s'))-q_2(s')P(s|s',\pi_2(s'))$$

$$\leq(q_1(s')-q_2(s'))P(s|s',\pi_1(s'))+q_2(s')P(s|s',\pi_1(s'))-q_2(s',\pi_1(s'))P(s|s',\pi_1(s'))$$

$$=(q_1(s')-q_2(s'))P(s|s',\pi_1(s'))+\left[\sum_{a\neq\pi_1(s')}q_2(s',a)\right]P(s|s',\pi_1(s'))$$

$$=(q_1(s')-q_2(s'))\sum_{a'}\pi_1(a'|s')P(s|s',a')+\sum_{a'}\pi_1(a'|s')\left[\sum_{a\neq\pi_1(s')}q_2(s',a)\right]P(s|s',a')$$

$$\leq\mathbb{I}_{s'}(q_1(s')-q_2(s'))\sum_{a'}\pi_1(a'|s')P(s|s',a')+\sum_{a'}g(s',a')P(s|s',a'),$$

where the third line follows from the fact that $-q_2(s')P(s|s',\pi_2(s'))\leq -q_2(s',\pi_1(s'))P(s|s',\pi_1(s'))$, which holds because the right-hand side is simply zero when $\pi_1(s')\neq\pi_2(s')$.

Combining these two cases proves the desired inequality for all states. Applying Lemma 18, we arrive at

$$f(s) \leq \sum_{k=0}^{k(s)-1} \sum_{s' \in S_k} \sum_{a' \in A} g(s', a') p^{\pi_1}(s|s', a').$$

By the same arguments used in the proof of Lemma 19, we have shown

$$\sum_{s \neq s_L} \sqrt{q_1(s)} - \sqrt{q_2(s)} \leq \sqrt{|S|L} \sqrt{\sum_{s \neq s_L} \sum_{a \in A} g(s, a)}.$$

Noticing that

$$\sum_{s \neq s_L} \sum_{a \in A} g(s, a) = \sum_{s \neq s_L} \left( \sum_{a \in A} \pi_1(a|s) \right) \left[ \sum_{a' \neq \pi_1(s)} q_2(s, a') \right] = \sum_{s \neq s_L} \sum_{a' \neq \pi_1(s)} q_2(s, a')$$

finishes the proof. $\qquad \square$

Finally, we are ready to prove Lemma 6.

*Proof of Lemma 6.* Recall the calculation at the beginning of this section:

$$\phi_H(\mathring{q}) - \phi_H(q_t) = \sum_{s \neq s_L} \sqrt{q_t(s)} h_s(\pi_t) - \sum_{s \neq s_L} \sqrt{\mathring{q}(s)} h_s(\mathring{\pi})$$

$$= \sum_{s \neq s_L} \sqrt{q_t(s)} \left( h_s(\pi_t) - h_s(\mathring{\pi}) \right) + (1 + \alpha(|A| - 1)) \sum_{s \neq s_L} \left( \sqrt{q_t(s)} - \sqrt{\mathring{q}(s)} \right).$$

Using Lemma 16 with $\pi_1 = \pi_t$ and $\pi_2 = \mathring{\pi}$, we bound the first term by $(1 + \alpha) \sum_{s \neq s_L} \sum_{a \neq \pi(s)} \sqrt{q_t(s, a)}$ for any mapping $\pi : S \to A$. Then we decompose the term $\sum_{s \neq s_L} \left( \sqrt{q_t(s)} - \sqrt{\mathring{q}(s)} \right)$ as

$$\sum_{s \neq s_L} \left( \sqrt{q_t(s)} - \sqrt{\mathring{q}(s)} \right) = \sum_{s \neq s_L} \left( \sqrt{q_t(s)} - \sqrt{q^{\pi}(s)} \right) + \sum_{s \neq s_L} \left( \sqrt{q^{\pi}(s)} - \sqrt{\mathring{q}(s)} \right)$$

$$\leq \sqrt{|S|L} \left( \sqrt{\sum_{s \neq s_L} \sum_{a \neq \pi(s)} q_t(s, a)} + \sqrt{\sum_{s \neq s_L} \sum_{a \neq \pi(s)} \mathring{q}(s, a)} \right)$$

$$\leq 2 \sqrt{|S|L \sum_{s \neq s_L} \sum_{a \neq \pi(s)} q_t(s, a) + \mathring{q}(s, a)}$$

where the second line is by using Lemma 19 with $\pi_1 = \pi_t$ and $\pi_2 = \pi$, and Lemma 20 with $\pi_1 = \pi$ and $\pi_2 = \mathring{\pi}$. This provides one upper bound for $\sum_{s \neq s_L} \left( \sqrt{q_t(s)} - \sqrt{\mathring{q}(s)} \right)$. On the other hand, it can also be trivially bounded as

$$\sum_{s \neq s_L} \sqrt{q_t(s)} - \sqrt{\mathring{q}(s)} \leq \sum_{s \neq s_L} \sqrt{q_t(s)} \leq \sqrt{|S|L}$$

where the second step uses Cauchy-Schwarz inequality and the fact $\sum_{s \neq s_L} q_t(s) = L$. Combining everything shows that $\phi_H(\mathring{q}) - \phi_H(q_t)$ is bounded by

$$(1 + \alpha) \sum_{s \neq s_L} \sum_{a \neq \pi(s)} \sqrt{q_t(s, a)} + (1 + \alpha|A|) \sqrt{|S|L} \min \left\{ 2 \sqrt{\sum_{s \neq s_L} \sum_{a \neq \pi(s)} q_t(s, a) + \mathring{q}(s, a)}, 1 \right\},$$

finishing the proof. $\qquad \square$

# D    Proof of Lemma 7

In this section, we provide the proof for Lemma 7. Throughout the section, we use the shorthand $H \triangleq \nabla^2 \phi_H(q_t)$. The key is clearly to analyze the Hessian inverse $H^{-1}$, which is done in Appendix D.1. With a better understanding of the Hessian inverse, we then finish the proof in Appendix D.2.

## D.1    Analyzing the Hessian inverse

To facilitate discussions, we first introduce some matrix notation. We see the Hessian $H$ as a matrix in $\mathbb{R}^{(|S||A|) \times (|S||A|)}$ in the natural way.[5] For subsets $D, E \subseteq S \times A$, we use $\mathbb{R}^{D \times E}$ to represent the set of matrices (in $\mathbb{R}^{|D| \times |E|}$ ) with the elements in $D$ indexing their rows and the elements in $E$ indexing their columns. The notation $M((s, a), (s', a'))$ represents the entry of a matrix $M$ in the row indexed by $(s, a)$ and in the column indexed by $(s', a')$. Let $U_k = \{(s, a) : s \in S_k, a \in A\}$ for $k = 0, \ldots, L - 1$, and $U_{j:k} = U_j \cup \cdots \cup U_k$. We use similar notations for vectors, and define $\mathbf{0}_U \in \mathbb{R}^U$ be the all-zero vector with elements in $U$ indexing its coordinates.

Define diagonal matrices $I_k, C_k, D_k \in \mathbb{R}^{U_k \times U_k}$ as

$$
\begin{aligned}
I_k &= diag\left\{1 : (s, a) \in U_k\right\}, \\
C_k &= diag\left\{c(s, a) : (s, a) \in U_k\right\}, \\
D_k &= diag\left\{d(s, a) : (s, a) \in U_k\right\},
\end{aligned}
$$

where

$$
c(s, a) = \frac{\alpha}{4(q(s) - q(s, a))^{3/2}} \quad \text{and} \quad d(s, a) = \frac{1}{4q(s, a)^{3/2}}.
$$

Also define transition matrices $P_k \in \mathbb{R}^{U_{k-1} \times U_k}$ such that $P_k((s, a), (s', a')) = P(s'|s, a)$, and $\widetilde{P}_k \in \mathbb{R}^{U_{0:k-1} \times U_k}$ such that $P_k((s, a), (s', a')) = P(s'|s, a)$ if $s' \in S_{k(s)+1}$ and $P_k((s, a), (s', a')) = 0$ otherwise.

Our first step is to write $H$ in a recursive way with the help of a sequence of matrices:

**Lemma 21.** *Define matrices $M_k \in \mathbb{R}^{U_{0:k} \times U_{0:k}}$ for $k = 0, \ldots, L - 1$ recursively as*

$$
M_k = \begin{pmatrix} M_{k-1} + \widetilde{P}_k C_k \widetilde{P}_k^\top & -\widetilde{P}_k C_k \\ -C_k \widetilde{P}_k^\top & C_k + D_k \end{pmatrix} = \begin{pmatrix} M_{k-1} & 0 \\ 0 & D_k \end{pmatrix} + \begin{pmatrix} \widetilde{P}_k \\ -I_t \end{pmatrix} C_k \begin{pmatrix} \widetilde{P}_k^\top & -I_t \end{pmatrix} \quad (24)
$$

*for $k = 1, \ldots, L - 1$, and $M_0 = C_0 + D_0$. Then we have $H = M_{L-1}$.*

*Proof.* The proof is by a direct verification based on the calculation of the Hessian done in Lemma 9. The claim is that $M_k$ consists of all the second-order derivatives with respect to $(s, a), (s', a') \in U_{0:k}$, but without the terms involving $P(s''|s, a)$ or $P(s''|s', a')$ for $s'' \in S_{k+1}$. To see this, note that this is clearly true for $M_0$ based on Lemma 9. Suppose this is true for $M_{k-1}$ and consider $M_k$.

We first show that the block $\widetilde{P}_k C_k \widetilde{P}_k^\top$ corresponds to Eq. (11) plus the last term of Eq. (12), that is

$$
\left(\widetilde{P}_k C_k \widetilde{P}_k^\top\right)((s, a), (s', a'))
$$

$$
= \sum_{(s_1, a_1) \in U_k} \sum_{(s_2, a_2) \in U_k} \widetilde{P}_k((s, a); (s_1, a_1)) \cdot C_k((s_1, a_1); (s_2, a_2)) \cdot \widetilde{P}_k((s', a'); (s_2, a_2))
$$

$$
= \sum_{(s_1, a_1) \in U_k} \widetilde{P}_k((s, a), (s_1, a_1)) \cdot C_k((s_1, a_1), (s_1, a_1)) \cdot \widetilde{P}_k((s', a'), (s_1, a_1))
$$

$$
+ \sum_{(s_1, a_1) \in U_k} \sum_{(s_2, a_2) \in U_k, (s_1, a_1) \neq (s_2, a_2)} \widetilde{P}_k((s, a), (s_1, a_1)) \cdot C_k((s_1, a_1), (s_2, a_2)) \cdot \widetilde{P}_k((s', a'), (s_1, a_1))
$$

$$
= \sum_{(s_1, a_1) \in U_k} P(s_1|s, a) \cdot \frac{\alpha}{4(q(s_1) - q(s_1, a_1))^{3/2}} \cdot P(s_1|s', a')
$$

$$
= \sum_{s'' \in S_k} \sum_{a'' \in A} \frac{\alpha P(s''|s, a) P(s''|s', a')}{4(q(s'') - q(s'', a''))^{3/2}}
$$

where the third step uses the fact that $C_k((s_1, a_1), (s_2, a_2)) = 0$ when $(s_1, a_1) \neq (s_2, a_2)$.

Then we verify that the blocks $-\widetilde{P}_k C_k$ and $-C_k \widetilde{P}_k^\top$ correspond to Eq. (10). Direct calculation shows that, for $(s, a) \in U_k$ and $(s', a') \in U_{k-1}$,

$$
\left(-\widetilde{P}_k C_k\right)((s', a'), (s, a)) = -\sum_{(s'', a'') \in U_k} \widetilde{P}_k((s', a'), (s'', a'')) C_k((s'', a''), (s, a))
$$

$$
= -\frac{\alpha}{4(q(s) - q(s, a))^{3/2}} \cdot P(s|s', a').
$$

Finally, the block $C_k + D_k$ corresponds to the first two terms of Eq. (12). This finishes the proof. $\square$

To study $H^{-1} = M_{L-1}^{-1}$, we next write $M_k^{-1}$ in terms of $M_{k-1}^{-1}$.

**Lemma 22.** *The inverse of $M_k$ (defined in Eq. (24)) is*

$$
M_k^{-1} = \begin{pmatrix} M_{k-1}^{-1} & 0 \\ 0 & D_k^{-1} \end{pmatrix} - \begin{pmatrix} M_{k-1}^{-1} \widetilde{P}_k \\ -D_k^{-1} \end{pmatrix} \left(C_k^{-1} + D_k^{-1} + \widetilde{P}_k^\top M_{t-1}^{-1} \widetilde{P}_k\right)^{-1} \left(\widetilde{P}_k^\top M_{k-1}^{-1} \quad -D_k^{-1}\right)
$$

$$
= \begin{pmatrix} M_{k-1}^{-1} - M_{k-1}^{-1} \widetilde{P}_k W_k \widetilde{P}_k^\top M_{k-1}^{-1} & M_{k-1}^{-1} \widetilde{P}_k W_k D_k^{-1} \\ D_k^{-1} W_k \widetilde{P}_k^\top M_{k-1}^{-1} & D_k^{-1} - D_k^{-1} W_k D_k^{-1} \end{pmatrix} \tag{25}
$$

*where $W_k = (C_k^{-1} + D_k^{-1} + \widetilde{P}_k^\top M_{k-1}^{-1} \widetilde{P}_k)^{-1}$.*

*Proof.* The first equality is by the Woodbury matrix identity

$$
(A + UBV)^{-1} = A^{-1} - A^{-1} U (B^{-1} + V A^{-1} U)^{-1} V A^{-1}
$$

and plugging in the definition of $M_k$ from Eq. (24) with

$$
A = \begin{pmatrix} M_{k-1} & 0 \\ 0 & D_k \end{pmatrix}, \quad U = V^\top = \begin{pmatrix} \widetilde{P}_k \\ -I_t \end{pmatrix}, \quad \text{and} \quad B = C_k.
$$

The second equality is by direct calculation. $\square$

The bottom right block of $M_k^{-1}$ plays a key role in the analysis, and is denoted by

$$
N_k = D_k^{-1} - D_k^{-1} W_k D_k^{-1} \tag{26}
$$

for $k = 1, \ldots, L - 1$, and $N_0 = M_0^{-1}$. The next lemma shows that we can focus on $N_k$ when analyzing specific quadratic forms of $H^{-1}$.

**Lemma 23.** *For any vector $w_k \in \mathbb{R}^{U_k}$, we have*

$$
\left(\mathbf{0}_{U_{0:k-1}}^\top, w_k^\top, \mathbf{0}_{U_{k+1:L-1}}^\top\right) H^{-1} \begin{pmatrix} \mathbf{0}_{U_{0:k-1}} \\ w_k \\ \mathbf{0}_{U_{k+1:L-1}} \end{pmatrix} \leq w_k^\top N_k w_k,
$$

*where $N_k$ is defined in Eq. (26).*

*Proof.* Based on Eq. (24) and the fact that $C_k$ is positive definite, we have

$$
M_k \succeq \begin{pmatrix} M_{k-1} & 0 \\ 0 & D_k \end{pmatrix}.
$$

Repeatedly using this fact, we can show

$$
H = M_{L-1} \succeq \begin{pmatrix} M_k & 0 & \cdots & 0 \\ 0 & D_{k+1} & \cdots & 0 \\ \vdots & \vdots & \ddots & \vdots \\ 0 & 0 & \cdots & D_{L-1} \end{pmatrix},
$$

and thus

$$H^{-1} = M_{L-1}^{-1} \preceq \begin{pmatrix} M_k^{-1} & 0 & \cdots & 0 \\ 0 & D_{k+1}^{-1} & \cdots & 0 \\ \vdots & \vdots & \ddots & \vdots \\ 0 & 0 & \cdots & D_{L-1}^{-1} \end{pmatrix}.$$

Note that for the last matrix above, the block with rows and columns indexed by elements in $U_k$ is exactly $N_k$, based on Lemma 22. Thus, taking the quadratic form on both sides with respect to the vector $\left(\mathbf{0}_{U_{0:k-1}}^\top, w_k^\top, \mathbf{0}_{U_{k+1:L-1}}^\top\right)$ finishes the proof. $\qquad\square$

Finally, we point out some important properties of $N_k$.

**Lemma 24.** *The matrix $N_k$ (defined in Eq. (26)) is positive definite and satisfies*

$$N_k \preceq D_k^{-1} \quad and \quad N_k \preceq C_k^{-1} + P_k^\top N_{k-1} P_k.$$

*Proof.* The fact that $N_k$ is positive definite is directly implied by Lemma 23. To prove the rest of the statement, we first apply Woodbury matrix identity to write $W_k$ as

$$W_k = (D_k^{-1} + C_k^{-1} + \widetilde{P}_k^\top M_{k-1}^{-1} \widetilde{P}_k)^{-1} = D_k - D_k(D_k + (C_k^{-1} + \widetilde{P}_k^\top M_{k-1}^{-1} \widetilde{P}_k)^{-1})^{-1} D_k$$
$$= D_k - D_k(D_k + (C_k^{-1} + P_k^\top N_{k-1}^{-1} P_k)^{-1})^{-1} D_k.$$

Plugging this back into the definition of $N_k$ gives:

$$N_k = D_k^{-1} - D_k^{-1} W_k D_k^{-1} = (D_k + (C_k^{-1} + P_k^\top N_{k-1}^{-1} P_k)^{-1})^{-1},$$

which shows $N_k \preceq D_k^{-1}$ and $N_k \preceq C_k^{-1} + P_k^\top N_{k-1} P_k$. $\qquad\square$

### D.2 Bounding the stability term

With the tools from the previous section, we are now ready to bound the stability term. We will use the following lemma to relate the quadratic form of $H^{-1}$ to only its diagonal entries.

**Lemma 25.** *If $M \in \mathbb{R}^{d \times d}$ is a positive semi-definite matrix, then for any $w \in \mathbb{R}^d$ with non-negative coordinates, we have*

$$w^\top M w \leq \left(\sum_{j=1}^d w(j)\right) \sum_{i=1}^d M(i,i)w(i).$$

*Proof.* Since $M$ is positive semi-definite, we have for any $i, j$, $(e_i - e_j)^\top M(e_i - e_j) \geq 0$, which implies

$$M(i,j) = M(j,i) \leq \frac{M(i,i) + M(j,j)}{2}.$$

Therefore,

$$w^\top M w = \sum_{i,j} w(i)M(i,j)w(j) \leq \frac{1}{2} \sum_{i,j} w(i)(M(i,i)+M(j,j))w(j) = \left(\sum_{j=1}^d w(j)\right) \sum_{i=1}^d M(i,i)w(i),$$

where we use the nonnegativity of $w$ in the second step as well. $\qquad\square$

We now bound $\|\widehat{\ell}_t\|_{H^{-1}}$ in terms of the diagonal entries of $N_k$.

**Lemma 26.** *Algorithm 1 guarantees*

$$\mathbb{E}\left[\left\|\widehat{\ell}_t\right\|_{H^{-1}}\right] \leq L\mathbb{E}\left[\sum_{k=0}^{L-1} \sum_{(s,a)\in U_k} \frac{N_k((s,a),(s,a))}{q_t(s,a)}\right].$$

*Proof.* Recall the definition of $\widehat{\ell}_t$: $\widehat{\ell}_t(s,a) = \frac{\ell_t(s,a)}{q_t(s,a)}\mathbb{I}\{s,a\}$ where we use the shorthand $\mathbb{I}\{s,a\} = \mathbb{I}\{s_{k(s)} = s, a_{k(s)} = a\}$. Therefore, we have

$$
\mathbb{E}\left[\left\|\widehat{\ell}_t\right\|_{H^{-1}}\right] = \mathbb{E}\left[\sum_{s,a}\sum_{s',a'}\frac{H^{-1}\left((s,a),(s',a')\right)}{q_t(s,a)q_t(s',a')}\mathbb{I}\{s,a\}\ell_t(s,a)\mathbb{I}\{s',a'\}\ell_t(s',a')\right]
$$
$$
\leq \mathbb{E}\left[\left(\sum_{s',a'}\mathbb{I}\{s',a'\}\ell_t(s',a')\right)\sum_{s,a}\left(\frac{H^{-1}\left((s,a),(s,a)\right)}{q_t(s,a)^2}\right)\mathbb{I}\{s,a\}\ell_t(s,a)\right],
$$

(27)

where in the last step we use Lemma 25 with $M$ being a matrix in the same shape of $H$ and with entry $M((s,a),(s',a')) = \frac{H^{-1}((s,a),(s',a'))}{q_t(s,a)q_t(s',a')}$ (which is clearly positive definite). Using the fact $\sum_{s',a'}\mathbb{I}\{s',a'\}\ell(s',a') \leq L$, $H^{-1}\left((s,a),(s,a)\right) \geq 0$, and $\ell_t(s,a) \in [0,1]$, we continue with

$$
\mathbb{E}\left[\left\|\widehat{\ell}_t\right\|_{H^{-1}}\right] \leq L\mathbb{E}\left[\sum_{s,a}\left(\frac{H^{-1}\left((s,a),(s,a)\right)}{q_t(s,a)^2}\right)\mathbb{I}\{s,a\}\right]
$$
$$
= L\mathbb{E}\left[\sum_{s,a}\frac{H^{-1}\left((s,a),(s,a)\right)}{q_t(s,a)}\right]
$$
$$
\leq L\mathbb{E}\left[\sum_{k=0}^{L-1}\sum_{(s,a)\in U_k}\frac{N_k((s,a),(s,a))}{q_t(s,a)}\right],
$$

where in the last step we use Lemma 23. $\qquad\square$

Next, we continue to bound the term involving $N_k$ using the following lemma.

**Lemma 27.** *Algorithm 1 guarantees*

$$
\sum_{k=0}^{L-1}\sum_{(s,a)\in U_k}\frac{N_k((s,a),(s,a))}{q_t(s,a)} \leq 8eL\left(\sqrt{L}+\frac{1}{\alpha L}\right)\sum_{s\neq s_L}\sum_{a\neq\pi(s)}\sqrt{q_t(s,a)}
$$

*for any mapping $\pi$ from $S$ to $A$.*

*Proof.* For notational convenience, define $R(s,a) = N_{k(s)}((s,a),(s,a))$. We first prove that for any $k = 1,\ldots,L-1$,

$$
\sum_{(s,a)\in U_k}\frac{R(s,a)}{q_t(s,a)} \leq 8\left(\sqrt{L}+\frac{1}{\alpha L}\right)\sum_{s\in S_k}\sum_{a\neq\pi(s)}\sqrt{q_t(s,a)} + \left(1+\frac{1}{L}\right)\sum_{(s',a')\in U_{k-1}}\frac{R(s',a')}{q_t(s',a')},
$$

(28)

and for $k = 0$,

$$
\sum_{a\in A}\frac{R(s_0,a)}{q_t(s_0,a)} \leq 8\left(\sqrt{L}+\frac{1}{\alpha L}\right)\sum_{a\neq\pi(s)}\sqrt{q(s_0,a)},
$$

(29)

where $\pi$ is any mapping from $S$ to $A$. Indeed, repeatedly applying (28) and using the fact $(1+^1/_L)^L \leq e$ show

$$
\sum_{(s,a)\in U_k}\frac{R(s,a)}{q_t(s,a)} \leq 8e\left(\sqrt{L}+\frac{1}{\alpha L}\right)\sum_{l=0}^{k}\sum_{s\in S_l}\sum_{a\neq\pi(s)}\sqrt{q_t(s,a)},
$$

and summing over $k$ finishes the proof.

To prove Eq. (29), note that by definition, when $s = s_0$ we have

$$
R(s,a) = \frac{4}{q_t(s,a)^{-3/2}+\alpha(q_t(s)-q_t(s,a))^{-3/2}} \leq 4\min\left\{q_t(s,a)^{3/2},\frac{(q_t(s)-q_t(s,a))^{3/2}}{\alpha}\right\}.
$$

Now consider two cases, if $\frac{q_t(s) - q_t(s, \pi(s))}{q_t(s, \pi(s))} \leq \frac{1}{L}$, then

$$\frac{R(s, \pi(s))}{q_t(s, \pi(s))} \leq \frac{4}{\alpha L} \sqrt{\sum_{a \neq \pi(s)} q_t(s, a)} \leq \frac{4}{\alpha L} \sum_{a \neq \pi(s)} \sqrt{q_t(s, a)}.$$

On the other hand, if $\frac{q_t(s) - q_t(s, \pi(s))}{q_t(s, \pi(s))} > \frac{1}{L}$, then $q_t(s, \pi(s)) \leq L(q_t(s) - q_t(s, \pi(s)))$ and

$$\frac{R(s, \pi(s))}{q_t(s, \pi(s))} \leq 4\sqrt{q_t(s, \pi(s))} \leq 4\sqrt{L}\sqrt{q_t(s) - q_t(s, \pi(s))} \leq 4\sqrt{L} \sum_{a \neq \pi(s)} \sqrt{q_t(s, a)}.$$

Combining the two cases and also the fact $\sum_{a \neq \pi(s)} \frac{R(s,a)}{q_t(s,a)} \leq 4 \sum_{a \neq \pi(s)} \sqrt{q_t(s, a)}$ proves Eq. (29).

It remains to prove Eq. (28). First, using the fact $N_k \preceq D_k^{-1}$ from Lemma 24, we again have

$$R(s, a) \leq 4q_t(s, a)^{3/2}. \tag{30}$$

At the same time, using another fact $N_k \preceq C_k^{-1} + P_k^\top N_{k-1} P_k$ from Lemma 24 and shorthand $R(s, a, s', a') \triangleq N_{k(s)}((s, a), (s', a'))$, we have

$$R(s, a) \leq \frac{4(q(s) - q(s, a))^{3/2}}{\alpha} + \sum_{(s_1, a_1), (s_2, a_2) \in U_{k-1}} P(s|s_1, a_1) P(s|s_2, a_2) R(s_1, a_1, s_2, a_2)$$

$$= \frac{4(q(s) - q(s, a))^{3/2}}{\alpha}$$
$$+ \sum_{(s_1, a_1), (s_2, a_2) \in U_{k-1}} P(s|s_1, a_1) q_t(s_1, a_1) P(s|s_2, a_2) q_t(s_2, a_2) \frac{R(s_1, a_1, s_2, a_2)}{q_t(s_1, a_1) q_t(s_2, a_2)}$$

$$\leq \frac{4(q(s) - q(s, a))^{3/2}}{\alpha}$$
$$+ \left( \sum_{(s_2, a_2) \in U_{k-1}} P(s|s_2, a_2) q_t(s_2, a_2) \right) \sum_{(s_1, a_1) \in U_{k-1}} P(s|s_1, a_1) q_t(s_1, a_1) \frac{R(s_1, a_1)}{q(s_1, a_1)^2}$$

$$= \frac{4(q(s) - q(s, a))^{3/2}}{\alpha} + q_t(s) \sum_{(s_1, a_1) \in U_{k-1}} P(s|s_1, a_1) \frac{R(s_1, a_1)}{q_t(s_1, a_1)},$$

where the second inequality is by applying Lemma 25 again, with $M \in \mathbb{R}^{U_{k-1} \times U_{k-1}}$ such that $M((s_1, a_1), (s_2, a_2)) = \frac{R(s_1, a_1, s_2, a_2)}{q_t(s_1, a_1) q_t(s_2, a_2)}$ (which is positive definite by Lemma 24). Again, we fix $s$ and consider two cases. First, if $\frac{q_t(s) - q_t(s, \pi(s))}{q_t(s, \pi(s))} \leq \frac{1}{L}$, then

$$\frac{R(s, \pi(s))}{q_t(s, \pi(s))} \leq \frac{4(q_t(s) - q_t(s, \pi(s)))^{3/2}}{\alpha q_t(s, \pi(s))} + \frac{q_t(s)}{q_t(s, \pi(s))} \sum_{(s_1, a_1) \in U_{k-1}} P(s|s_1, a_1) \frac{R(s_1, a_1)}{q_t(s_1, a_1)}$$

$$\leq \frac{4(q_t(s) - q_t(s, \pi(s)))}{\alpha q_t(s, \pi(s))} \sum_{a \neq \pi(s)} \sqrt{q_t(s, a)} + \left(1 + \frac{1}{L}\right) \sum_{(s_1, a_1) \in U_{k-1}} P(s|s_1, a_1) \frac{R(s_1, a_1)}{q_t(s_1, a_1)}$$

$$\leq \frac{4}{\alpha L} \sum_{a \neq \pi(s)} \sqrt{q_t(s, a)} + \left(1 + \frac{1}{L}\right) \sum_{(s_1, a_1) \in U_{k-1}} P(s|s_1, a_1) \frac{R(s_1, a_1)}{q_t(s_1, a_1)}.$$

On the other hand, if $\frac{q_t(s) - q_t(s, \pi(s))}{q_t(s, \pi(s))} > \frac{1}{L}$ and thus $q_t(s, \pi(s)) \leq L(q_t(s) - q_t(s, \pi(s)))$, then using Eq. (30) we have

$$\frac{R(s, \pi(s))}{q(s, \pi(s))} \leq 4\sqrt{q(s, \pi(s))} \leq 4\sqrt{L \sum_{a \neq \pi(s)} q_t(s, a)} \leq 4\sqrt{L} \sum_{a \neq \pi(s)} \sqrt{q_t(s, a)}.$$

Combining the two cases and also $\sum_{a\neq\pi(s)}\frac{R(s,a)}{q_t(s,a)}\leq 4\sum_{a\neq\pi(s)}\sqrt{q_t(s,a)}$ (using Eq. (30) again) leads to

$$\sum_a \frac{R(s,a)}{q_t(s,a)} \leq 8\left(\sqrt{L}+\frac{1}{\alpha L}\right)\sum_{a\neq\pi(s)}\sqrt{q_t(s,a)}+\left(1+\frac{1}{L}\right)\sum_{(s_1,a_1)\in U_{k-1}}P(s|s_1,a_1)\frac{R(s_1,a_1)}{q_t(s_1,a_1)}.$$

Finally, we sum over all $s\in S_k$ and obtain

$$\sum_{(s,a)\in U_k}\frac{R(s,a)}{q_t(s,a)}$$
$$\leq 8\left(\sqrt{L}+\frac{1}{\alpha L}\right)\sum_{s\in S_k}\sum_{a\neq\pi(s)}\sqrt{q_t(s,a)}+\left(1+\frac{1}{L}\right)\sum_{s\in S_k}\sum_{(s',a')\in U_{k-1}}P(s|s',a')\frac{R(s',a')}{q_t(s',a')}$$
$$= 8\left(\sqrt{L}+\frac{1}{\alpha L}\right)\sum_{s\in S_k}\sum_{a\neq\pi(s)}\sqrt{q_t(s,a)}+\left(1+\frac{1}{L}\right)\sum_{(s',a')\in U_{k-1}}\frac{R(s',a')}{q_t(s',a')}.$$

This proves Eq. (28) and thus finishes the proof. $\qquad\square$

We are now ready to finish the proof for Lemma 7.

*Proof of Lemma 7.* Combining Lemma 26 and Lemma 27, we prove

$$\mathbb{E}\left[\|\widehat{\ell}_t\|^2_{\nabla^{-2}\phi_H(q_t)}\right]\leq\mathbb{E}\left[8eL^2\left(\sqrt{L}+\frac{1}{\alpha\cdot L}\right)\sum_{s\neq s_L}\sum_{a\neq\pi(a)}\sqrt{q_t(s,a)}\right].$$

It thus remains to prove the other bound

$$\mathbb{E}\left[\|\widehat{\ell}_t\|^2_{\nabla^{-2}\phi_H(q_t)}\right]\leq 4\sqrt{L|S||A|}.$$

This is simply by considering only the regular $1/2$-Tsallis entropy part of the regularizer: $\phi_D(q)=-\sum_{s,a}\sqrt{q(s,a)}$. Specifically, by Lemma 9 we have $\nabla^2\phi_H(q_t)\succeq\nabla^2\phi_D(q_t)$, and thus

$$\mathbb{E}\left[\left\|\widehat{\ell}_t\right\|^2_{\nabla^{-2}\phi_H(q_t)}\right]\leq\mathbb{E}\left[\left\|\widehat{\ell}_t\right\|^2_{\nabla^{-2}\phi_D(q_t)}\right]$$
$$=\mathbb{E}\left[\sum_{s,a}\frac{4q_t(s,a)^{3/2}}{q_t(s,a)^2}\mathbb{I}\{s,a\}\ell_t(s,a)^2\right]$$
$$\leq 4\mathbb{E}\left[\sum_{s,a}\sqrt{q_t(s,a)}\right]$$
$$\leq 4\sqrt{L|S||A|},$$

where the last step uses the Cauchy-Schwarz inequality. $\qquad\square$