[Reviews · NeurIPS 2020]

Review 1

Summary and Contributions: The paper addresses the problem of learning MDPs in the episodic, finite horizon setting with known transition kernel in the bandit setting, under both stochastic and adversarially generated rewards. The main contribution of the paper is a "best of both worlds" algorithm which works under both stochastic and adversarially generated rewards, without needing to know the setting a-priori, and an analysis which includes optimistic rates in the stochastic setting close to the best known rates in prior work. ------------------------------- I have read the rebuttal and other reviews. In the section for additional feedback and comments I am providing most places where I was initially confused or it took me some time to figure out the derivation. Regarding novelty of the regularizer -- I agree that the regularizer does not appear explicitly in prior work, however, the ideas (modulo adding the log-barrier, which I still think is there due to authors following results from prior work i.e. showing \tilde q_t is close to q_t, rather than being necessary) follow rather naturally. I do not think this should be considered as a major contribution of this work. I am happy with the response regarding the gap_\min and somewhat satisfied with the comment about implementation. I am raising my current score to 7.

Strengths: Soundness of the claims: This work is theoretical. All the proofs of main results can be found in the supplementary material and seem sound. Significance and novelty of the contributions: The core novel contribution of this work is the technical analysis of the proposed algorithm. The increased difficulty of the analysis compared to prior (Zimmert and Seldin 2019, Zimmert et al. 2019) comes from the fact that the "stability" term in the analysis of FTRL now contains a non-diagonal Hessian and it is harder to extract the correct bound for the "penalty" term, which is only in terms of occupancy measures with sub-optimal value functions. The key lemmas are Lemma 6 and Lemma 7 in the supplementary material. Relevance to the NeurIPS community: The analysis can be of interest to other researchers working on problems in Bandit algorithms and Online learning. The results might be of interest to researchers working on more theoretical RL problems.

Weaknesses: Soundness of the claims: While the general ideas seem somewhat clear, most of the proofs read more like sketches and some of the claims need to be verified by hand. For example in the proof of Lemma 9 the authors do not provide a derivation for the each of the elements of the Hessian but merely state that the result follows from a direct computation (which seems to be left as an exercise to the reader). Other examples where more details would not hurt are the proof of Lemma 13, where the authors bound \|q_t - \tilde \q_t\| in a self-bounding way, to achieve the upper bound but no mention of this is given other than the final result and the proof of Lemma 27, where derivations on lines 720-722 were not entirely straightforward. Overall the Appendix can benefit from more details in the proofs. Significance and novelty of contribution: The core ideas of this work are not novel. The fundamental ideas which the work builds on and combines are reducing the learning MDP problem to essentially a bandit learning problem by using occupancy measures introduced in (Zimin and Neu 2013) and the analysis of FTRL algorithms for stochastic bandits by using the self-bounded property of regret in such algorithms introduced in (Wei and Luo 2018) and extended in (Zimmert and Seldin 2019, Zimmert et al. 2019). In particular, the algorithm is essentially FTRL with the regularizer in (Zimmert et al. 2019), which seems to be needed to induce the correct dependence on gaps, on the occupancy measures. Further the algorithm also needs to mix in a log-barrier regularizer, however, it is not clear if this is really necessary or it just arises as a byproduct of the techniques used in the analysis. To the best of my understanding the gap_min factor in the bound comes exactly from the use of log-barrier. While such a factor is unavoidable for optimistic algorithms as shown in (Simchowitz and Jamieson 2019) this is due to the UCBs for each of the reward functions. Given that the FTRL algorithm works with occupancy measures which seems to be equivalent to directly working with policies, it is not at all clear if this factor is necessary in the bound as the problem of optimistic estimators for each reward function does not arise. Relevance to the NeurIPS community: This work will probably not be beneficial to researchers working on the more empirical side of RL. Further, the authors do not give any details about how to implement the optimization problem in the FTRL step. Given that a non-trivial portion of (Zimin and Neu 2013) was dedicated to solving the optimization problem in the case of the relative entropy regularizer and at least in the bandits literature Tsallis entropy regularizers actually require more work to compute the necessary normalization constant it would have been nice to at least have a short discussion about carrying out the FTRL step.

Correctness: Theoretical claims are most likely correct. I could not check all of the proofs in detail, however.

Clarity: The paper is mostly well written, however, the supplementary needs more work.

Relation to Prior Work: Relation to prior work is clearly discussed and novel contributions are clear.

Reproducibility: Yes

Additional Feedback: Post rebuttal comments: First I would recommend the authors include a section in the appendix with most definitions and notation. This would greatly help with reading the proofs as I continuously had to track back and find notation introduced on the go. A page similar to the one found in the work of Simchowitz and Jamieson 2019 in the beginning of the appendix would be extremely useful. Detailed comments: Finishing the proof of Theorem on lines 432-433 -- please give another equation where you actually combine all of the terms. I could not verify the final steps and correct dependence on the introduced X,Y,Z before having to write this for myself. Lines 444-445 and 447-448 -- please add an extra step before applying AM-GM where you introduce all the extra terms in the form of the gaps and z. Line 516 -- please recall the definition of \psi_t here, it would make life easier as one would not need to search again for the fact that \psi_t decomposes as a sum of \phi_H and \phi_L. Lines 522-527 -- I am not entirely sure I follow this proof. It seems like the authors want to argue by contradiction showing that for any q' not satisfying the norm inequality, q' would not be a minimizer of G_t. I would see how this leads to the wanted conclusion if the assumption was \|q' - q_t\| \geq 1. Lines 528-529 -- What earlier argument is being repeated here. It doesn't look like the authors have yet shown anything regarding 1/2 q_t(s,a) \leq q'(s,a) \leq 2q_t(s,a) as the norm inequality is yet to be verified. It's verification seems to follow only after in lines 532-533. Line 543 -- is there a typo where q should actually be q_t? Lines 555-556 -- In the statement of the Lemma notation for q is overloaded as it also appears in the argmin for the definition of q_1. Please change this. Proof of Lemma 21 -- instead of just stating in a short paragraph how the derivation follows, please actually provide details, i.e., replace lines 670-672 with actual display equations. Proof of Lemma 22 -- actually give the proof with a statement of the Woodbury inversion formula rather than only state how it follows. Finally, I do not know if other reviewers had problems with the hyperlinks in the document, but for me the hyperlinks did not work.


Review 2

Summary and Contributions: This paper addresses the problem that if the best-of-both-the-worlds algorithm exists in episodic MDPs when the transition matrix is known. Inspired by [27, 29], which study the analogs in MAB, the authors derive the self-bounding regret by using a hybrid regularizer. However, non-diagonal Hessian is admitted when analyzing the non-decomposable regularizer. The contribution is to bound the stability term recursively based on the layered structure. As a result, the proposed algorithm attains O(\sqrt{L|S||A|T}+|S||A|Llog(T) ) and O(log T) against the adversarial environment and the stochastic environment respectively.

Strengths: To the best of my knowledge, this paper is the first work adaptively provides regret guarantees in Episodic MDPs. It sheds light on designing a single robust algorithm in more complicated reinforcement learning problems.

Weaknesses: Even with the strong assumption that the transition matrix is known, the algorithm does NOT attain optimal regret in the adversarial setting (In [23], the regret of their algorithm can be bounded by 2\sqrt{L|S||A|T} without the prior knowledge of the transition matrix). In particular, when |S||A|L is a large number, the second term in the upper bound proposed by this paper will dominate. Hence, it still remains a mystery in episodic MDPs when the knowledge of the environment is unavailable. Furthermore, the bound author tends to match is the worst-case lower bound not specifice to P. When the problems are complicated, worst-case lower bound is not meaningful to depict the problem.

Correctness: Most of the proofs and methods are reasonable.

Clarity: The main text is clear and easy to follow but the appendix would be unfriendly for the reader who is unfamiliar with the related works.

Relation to Prior Work: The author clearly compares their work with the previous works.

Reproducibility: Yes

Additional Feedback: Although the method to analyze the regularizer is novel, it would be better to know what else the analysis can be employed in various ways. _____After rebuttal_________________________ Thanks for the author's response. I know that I misunderstand the bounds in the previous works. However, although the authors do an excellent job of presenting their results in the main text, I still doubt their proofs in the appendix. Also, I would like to emphasize that this work only tends to match the worst-case lower bound. When the problem is complicated, to match the worst-case lower bound is much easier than the problem-specific bound. At least, I hope the future works can involve the transition matrix in the upper(lower) bound. For these reasons, I will update my score to 6 but not anymore.


Review 3

Summary and Contributions: This paper studies learning in the the episodic MDP setting with known transitions and shows that follow-the-regularized-leader with a specially designed regularizer (built on Tsallis entropy) allows a "best-of-both-worlds" bound that, simultaneously and without knowledge of the setting, has a sqrt(T) regret in the adversarial case and a log(T) is the stochastic case. Furthermore, the regret bound has a sqrt(C) term, where C is a bound on the total adversarial corruption of the losses, which allows it to interpolate between the two cases smoothly. Technically, the paper analysis the episodic MDP problem as a linear bandit problem on the space of occupancy measures and extends the analysis to the MDP setting. Non-trivial technical upgrades are needed to analyze the FTRL algorithm. post rebuttal: I remain supportive of the paper.

Strengths: The best-of-both-worlds problem is certainly worth exploring and is relevant to the community. While episodic MDPs are well understood, the best-of-both-worlds problem hasn't been explored in this context (even though it has been addressed for MAB). Even though the paper only considers known transitions, I still think the problem is important. The results are nice and complete; the best-of-both-worlds question is satisfactorily answered with an affirmative. The paper is well written, easy to follow, and clearly explains all the technical contributions while providing necessary context. Technically, the regularizer design is very interesting and the analysis tools developed could be of independent interest. It seems like a strong contribution.

Weaknesses: The contributions are purely theoretical, as the algorithm may be difficult to implement. Specifically, computing argmins over \Omega involves a convex optimization over a very complicate polytope, which seems infeasible except over small problems. As such, no empirical evaluations are given.

Correctness: Yes

Clarity: Yes

Relation to Prior Work: Yes, the literature review is comprehensive.

Reproducibility: Yes

Additional Feedback:


Review 4

Summary and Contributions: The paper considers the best-of-both-world problem for the episodic mdps, for bandit feedback setting with known Markov transition function. The one-step loss function may change per episode can be either be stochastic or adversarial. The authors consider the framework of FTRL with a newly designed hybrid regularizer (a log-barrier regularizer and a modified Tsallis entropy regularizer) to achieve O(\sqrt{T}) regret for the adversarial setting and log(T) regret for the stochastic setting.

Strengths: The authors follow the research line of work of applying online learning approach to study episodic MDP problems and further uses generalized FTRL approach with a novel hybrid regularizer. The work is technical and uses previous proof ideas for best-of-both-worlds problem from (Zimmert & Seldin 2018), (Zimmert et al. 2019) to achieve a generic form for regret guarantee and then specialize to both settings. One particular technical difficulty is that the hybrid regularizer has non-diagonal Hessian for which the authors make contribution to its analysis through the structure of the MDP transition.

Weaknesses: Algorithm efficiency: I was wondering if the update rule for the generalized FTRL with the hybrid regularizer may be efficiently implemented or may not have a semi-closed form expression similar to O-REPS or UC-O-REPS.

Correctness: The authors feedback has cleared the doubts of the reviewer. The proofs are read through and most checked at least once. (except for A.2 and the performance difference lemma [Lemma 5.2.1 in Kakade, 2003]).

Clarity: The authors has given very detailed proofs for the regret derivation, while the high-level outline of proof techniques is given In the main text. The paper is very well written and provides many analytical ideas along the way.

Relation to Prior Work: The paper has discussed its relations to current separate results on log regret bound in the stochastic setting and \sqrt{T} regret bound for the adversarial case. To the author's best knowledge it is the first result to achieve best of both world in the episodic MDP setting with known dynamic. (although with a slightly worse dependency on other problem instance parameters when using a single algorithm with fixed hyper-parameters)

Reproducibility: Yes

Additional Feedback: Q1. In line 523, the authors assume that ||q’-q_t||_H =1, do we need to consider the case ||q’-q_t||_H>1 as well? Q2. How is the first inequality in line 629 derived? Could the authors reply on the above questions as well as a verification of the removal of the expectation operator in the upper bound for the stability term (If true, what would the change be in the stochastic setting.) Thank you. For Q1: By the authors' explanations (as well as the reference [Lee et al. 2020]): in line 522-523, the goal is to prove that ||\tilde{q}_t-q_t||\leq 1. To prove this the authors showed that if ||q'-q_t||_H=1, then G_t(q') \geq G_t(q_t). Since G_t is convex: thus the sub-level set S:=\{q: G_t(q)\leq G_t(q_t)\} is convex. Also, \tilde{q}_t is in S by optimality of \tilde{q}_t. S is contained in \{q: ||q'-q_t||_H \leq 1\}. Thus ||\tilde{q}_q-q_t||_H \leq 1. For Q2: the derivation in line 629 holds true as explained in the authors feedback. Third, the expectations in line 709 and 740 involve the randomness up to time t (including t), the authors reduce (q(s,a))^2 to q(s,a) by taking conditional expectation at time t only. The expectations shown at the end of the proof then only involve randomness before time t. (Maybe it's better to that the authors add a little remark on this. Thanks) minor typos: line 543-544: missing subscripts: 2q_t(s,a). line 472 in expression (9): missing a prime for a.

[Author Response · NeurIPS 2020]

We thank all reviewers for their valuable comments! Below we address the issues raised by each reviewer.

**R2:   "most of the proofs read more like sketches".** We thank the reviewer for pointing out the two places where
more derivations are helpful and will revise accordingly. However, we respectfully disagree with the claim that *most*
*proofs are like sketches* — indeed, our proof is 21 pages long, and R7 agrees that we have "given very detailed proofs
for the regret derivation".

**"The core ideas of this work are not novel ... the algorithm is essentially FTRL with the regularizer in Zimmert**
**et al. 2019..."** In fact, as we mentioned in the last paragraph of Sec 2, the regularizer $-\sum_a(\sqrt{q(a)} + \sqrt{1 - q(a)})$ for
multi-armed bandits is not even explicitly mentioned or analyzed in Zimmert et al. 2019, let alone the extra modification
we need to do for MDPs (replacing 1 by $q(s) = \sum_a q(s,a)$ for state $s$). Also note that a direct extension of Zimmert
and Seldin, 2019 does not work as we argue in the second paragraph of Sec 3, implying that careful thinking is needed
to extend the ideas to MDPs. R6 also finds our "regularizer design very interesting". Algorithmic novelty aside, our
analysis also contains many novel ideas, as the reviewer agrees (in the "Strengths" section).

**"$\Delta_{\text{MIN}}$ factor in the bound comes exactly from the use of log-barrier ... not at all clear if this factor is necessary."**
To clarify, the $\Delta_{\text{MIN}}$ factor *does not* come from the log-barrier; instead, it comes from the second term is the penalty
bound stated in Lemma 6, which is only about the Tsallis entropy part. It is indeed unclear if this factor is necessary, but
to our knowledge, no existing algorithms, optimistic or not, enjoy a logarithmic regret bound without this dependence.

**"how to implement the optimization problem in the FTRL step."** Since this is a convex problem with $\mathcal{O}(L+|S||A|)$
linear constraints, one can apply any standard convex solver to implement the algorithm (accuracy $1/T$ is enough
clearly).

**R3:**   There seem to be quite some misunderstandings in the "Weaknesses" section, and we are not sure we fully
understand all comments. We try our best to clarify below and sincerely hope that the paper can be re-evaluated.

**"the algorithm does NOT attain optimal regret in the adversarial setting ... when $|S||A|L$ is a large number, the**
**second term in the upper bound proposed by this paper will dominate."** Our bound $\widetilde{\mathcal{O}}(\sqrt{L|S||A|T} + L|S||A|)$
*is* order-optimal (up to log terms). Please note that the optimal bound $\Theta(\sqrt{L|S||A|T})$ is only meaningful when
$L|S||A| \leq T$ (since otherwise the regret is linear), and in this regime, the second term $L|S||A|$ in our upper bound is
dominated by the first term. So indeed, our upper bound is simply $\widetilde{\mathcal{O}}(\sqrt{L|S||A|T})$, which is optimal.

**"$0.04\sqrt{L|S||A|T}$ is the minimax lower bound when $P$ is unknown."** We assume that this specific bound is taken
from Zimin and Neu, 2013 (Sec 5), which is a lower bound *even when $P$ is known* (the entire paper of Zimin and Neu,
2013 is about known transition).

**"In [23], the regret of their algorithm can be bounded by $2\sqrt{L|S||A|T}$ without the prior knowledge of the**
**transition matrix."** We are not sure we understand the comment — the work [23] (Simchowitz and Jamieson) is only
for stochastic losses. For the adversarial case with unknown transition and bandit feedback, the best lower and upper
bounds are $\Omega(L\sqrt{|S||A|T})$ and $\widetilde{\mathcal{O}}(L|S|\sqrt{|A|T})$ respectively as shown in Jin et al., 2020.

**R6:**   Please see the last response to R2 on the implementation issue.

**R7:**   Please see the last response to R2 on the implementation issue. Due to the complicated nature of the regularizer,
our algorithm does not admit a "semi-closed form expression" unfortunately, but note that even though UC-O-REPS
or UOB-REPS admits such a semi-closed form, it still requires solving a convex problem with as many positivity
constraints (O-REPS, on the other hand, only requires solving an unconstrained convex problem).

**"In Lemma 26, in line 709 ... should the expectation be removed?"** No, the expectation should stay. This is because
the expectation here is with respect to everything up to episode $t$ (including episode $t$). What we do in line 709 is merely
to first take the conditional expectation with respect to the randomness in episode $t$ alone (so that $\mathbb{E}_t[\mathbb{I}\{s,a\}] = q_t(s,a)$),
and after that, the expectation with respect to the past remains.

**"do we need to consider the case $\|q' - q_t\|_H > 1$ as well?"** No. If we can show that for all $\|q' - q_t\|_H = 1$,
$G_t(q') \geq G_t(q_t)$ holds, then by convexity and the optimality of $\tilde{q}_t$, this implies $\|\tilde{q}_t - q_t\|_H \leq 1$, which is all we need.
The same argument can be found in several earlier works, such as Lemma 9 of Lee et al. 2020.

**"How is the first inequality in line 629 derived?"** The inequality is equivalent to $-q_2(s')P(s|s', \pi_2(s')) \leq$
$-q_2(s', \pi_1(s'))P(s|s', \pi_1(s'))$, which holds because the right-hand side is simply zero (to see this, note that
$q_2(s', \pi_1(s')) = 0$ because $\pi_2$ is a deterministic policy that picks action $\pi_2(s')$ not equal to $\pi_1(s')$ at state $s'$).
We will clarify this in the next version. Thanks for the question.

[Meta-Review · NeurIPS 2020]

This paper was well-received by the reviewers and the author response was effective in addressing the concerns raised in the initial reviews. As a result, several reviewers updated their scores. The paper is clearly suitable for publication without significant changes.